# SELF-ADAPTIVE RETRIEVAL-AUGMENTED REINFORCE-MENT LEARNING FOR TIME SERIES FORECASTING

## ABSTRACT

Deep learning models for time series forecasting, typically optimized with Mean Squared Error (MSE), often exhibit spectral bias. This phenomenon arises because MSE prioritizes minimizing errors in high-energy, typically low-frequency components, leading to an underfitting of crucial, lower-energy high-frequency dynamics and resulting in overly smooth predictions. To address this, we propose Self-adaptive Retrieval-augmented Reinforcement learning for time series Forecasting (SRRF), a novel plug-and-play training enhancement. SRRF uniquely internalizes high-frequency modeling capabilities into base models during training, ensuring no additional inference costs or architectural changes for the base model. The framework operates by first employing Retrieval-Augmented Generation (RAG) to provide contextual grounding via relevant historical exemplars. Subsequently, building on this contextual guidance, a Reinforcement Learning (RL) agent learns an adaptive policy to correct and enhance initial forecasts, optimized via a reward function that promotes both overall predictive accuracy and fidelity to high-frequency details. Comprehensive evaluations on diverse benchmarks demonstrate that models trained with the SRRF methodology substantially improve upon their original versions and other state-of-the-art techniques, especially in accurately predicting volatile series and fine-grained temporal patterns. Qualitative and spectral analyses further confirm SRRF's effectiveness in mitigating spectral bias and enhancing high-frequency representation. Our code is available at https://anonymous.4open.science/r/ACAC-9999/README.md.

## 1 INTRODUCTION

Accurate time series forecasting is a pivotal task across diverse domains, from optimizing energy grids and supply chains to financial modeling and climate science (Hyndman & Athanasopoulos, 2018). The past decade has witnessed significant advancements driven by deep learning, with a proliferation of sophisticated architectures based on Convolutional Neural Networks (CNNs) (donghao & wang xue, 2024), Recurrent Neural Networks (RNNs) (Hochreiter & Schmidhuber, 1997), and Transformers (Vaswani et al., 2017). These models have continually pushed the state-of-the-art by effectively modeling complex temporal dependencies.

Despite these architectural innovations, a fundamental limitation persists, rooted in the near-ubiquitous use of the **Mean Squared Error** (MSE) loss as the training objective. This choice, while computationally convenient, induces systemic flaws that severely limit the high-fidelity prediction capabilities of even the most advanced models. The primary flaw is **spectral bias** (Xu et al., 2019). In most real-world time series, low-frequency components (e.g., trends, seasonality) dominate the signal's energy. The MSE loss, being proportional to the squared error, forces gradient-based optimizers to disproportionately prioritize fitting these high-energy components to achieve a rapid reduction in the overall loss. This process effectively transforms the deep learning model into an implicit low-pass filter, systematically underfitting the crucial, albeit low-energy, high-frequency details. A closely related second flaw is **amplitude suppression**. In their effort to minimize squared errors across an entire dataset, models learn a conservative, risk-averse mapping that regresses towards the mean. This results in predictions that are confined to a narrower value range than the true signal, consistently failing to forecast the magnitude of critical extreme events.

These two flaws are not merely theoretical concerns; they represent a critical barrier to practical deployment. The failure of models to capture high-frequency details and predict extreme values is precisely what makes them unreliable in high-stakes applications. This systemic failure is illustrated in Figure 1. On the one hand, a standard model completely ignores the high-frequency oscillations of a signal, regressing to the mean trend. On the other hand, it systematically underestimates the signal's volatility, failing to predict sharp peaks and troughs.

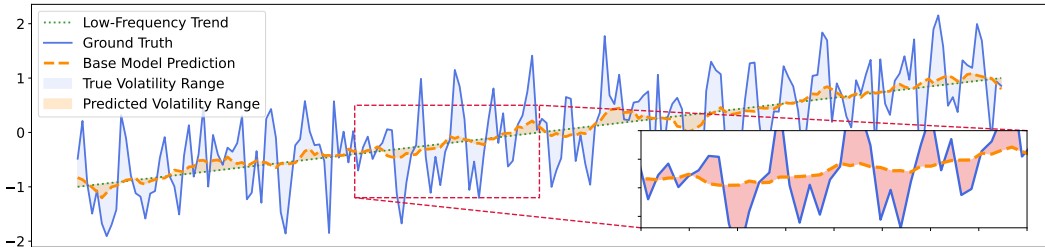

Figure 1: Illustration of two fundamental flaws common in standard forecasting models, using synthetic data for a clear depiction. **(a) Spectral Bias:** A base model (dashed orange) ignores the high-frequency oscillations of the ground truth (solid blue), learning only the underlying trend. The zoomed-in view highlights this failure. **(b) Amplitude Suppression:** The predicted volatility envelope (orange) is systematically compressed compared to the true volatility (blue), demonstrating the failure to forecast extreme values.

To overcome this dual challenge, we propose the Self-adaptive Retrieval-augmented Reinforcement learning framework (SRRF). SRRF is a novel, plug-and-play *training enhancement* that addresses the root cause of the problem without altering the base model's architecture. It integrates Reinforcement Learning (RL) with Retrieval-Augmented Generation (RAG) to create a sophisticated correction mechanism. The RAG component provides crucial context by retrieving historical exemplars of complex dynamics, while the RL agent learns an adaptive policy to refine the base model's initial forecast. This policy is optimized via a reward function that is explicitly sensitive to both overall accuracy and high-fidelity details, thus escaping the limitations of the MSE gradient. As SRRF is a training-time methodology, it internalizes these enhanced capabilities into the base model, incurring **no additional overhead at inference time**.

Our contributions are summarized as follows:

- We identify and characterize two systemic flaws in MSE-trained models—spectral bias and amplitude suppression—and propose a novel framework, SRRF, designed as a plug-and-play training enhancement to counteract them.

- We provide a rigorous, self-contained theoretical justification for SRRF in the main text, demonstrating from both an optimization and a statistical perspective how it resolves the gradient dilemma of standard losses and manages the bias-variance trade-off.

- We present comprehensive empirical results demonstrating that SRRF yields significant performance improvements across a wide range of state-of-the-art models. Crucially, we provide a head-to-head comparison against simpler differentiable alternatives, verifying the necessity of our RL-based decoupled design.

## 2 PROBLEM FORMULATION AND THEORETICAL MOTIVATION

In this section, we establish the theoretical foundations that motivate our work. We first provide a formal problem definition for time series forecasting. We then deconstruct the systemic flaws of standard training paradigms from two complementary perspectives: an optimization view, which reveals a fundamental **gradient dilemma** that hinders the learning of fine details, and a statistical view, which frames the problem as a **bias-variance trade-off** in forecast correction. This analysis provides a self-contained justification for the architectural principles of our proposed framework. For interested readers, detailed mathematical derivations are provided in Appendix B.

## 2.1 Formal Problem Definition

Let a multivariate time series be denoted as a sequence of observations $\mathcal{X} = (\mathbf{x}_1, \mathbf{x}_2, \ldots, \mathbf{x}_t, \ldots)$, where each observation $\mathbf{x}_t \in \mathbb{R}^D$ is a $D$-dimensional vector. The task of time series forecasting is, given a lookback window of $L$ historical observations, $\mathbf{X}_t = (\mathbf{x}_{t-L+1}, \ldots, \mathbf{x}_t) \in \mathbb{R}^{L \times D}$, to predict the corresponding future sequence of $P$ observations, $\mathbf{Y}_t = (\mathbf{x}_{t+1}, \ldots, \mathbf{x}_{t+P}) \in \mathbb{R}^{P \times D}$. A deep forecasting model, parameterized by $\theta$, learns a mapping $f_\theta : \mathbb{R}^{L \times D} \to \mathbb{R}^{P \times D}$ that generates a prediction $\hat{\mathbf{Y}}_t = f_\theta(\mathbf{X}_t)$. The parameters $\theta$ are typically optimized by minimizing a loss function $\mathcal{L}(\hat{\mathbf{Y}}_t, \mathbf{Y}_t)$ over a training dataset.

## 2.2 The Optimization View: A Gradient Dilemma

The choice of the loss function $\mathcal{L}$ profoundly impacts the learning dynamics. We argue that standard loss functions, such as Mean Squared Error (MSE) and Mean Absolute Error (MAE), create an irreconcilable **gradient dilemma** for high-fidelity forecasting.

**The Vanishing Gradient Problem of MSE.** The MSE loss, $\mathcal{L}_{MSE} = \frac{1}{P \cdot D} \sum_{i,j} (y_{ij} - \hat{y}_{ij})^2$, provides a gradient with respect to a prediction $\hat{y}$ that is directly proportional to the error, i.e., $\nabla_{\hat{y}} \mathcal{L}_{MSE} \propto (y - \hat{y})$. While this ensures smooth convergence when errors are large, it becomes the root cause of spectral bias and amplitude suppression. For small errors, which are characteristic of the fine-grained, high-frequency details or subtle amplitude variations that a model has almost captured, the gradient signal becomes vanishingly small. Consequently, the optimizer lacks the necessary signal to perform the final, precise adjustments required for high-fidelity prediction, leading to overly smooth forecasts that neglect these critical details.

**The Instability of MAE.** The MAE loss, $\mathcal{L}_{MAE} = \frac{1}{P \cdot D} \sum_{i,j} |y_{ij} - \hat{y}_{ij}|$, attempts to solve this with a constant-magnitude gradient, $\nabla_{\hat{y}} \mathcal{L}_{MAE} \propto \text{sgn}(y - \hat{y})$. This ensures a strong, non-vanishing learning signal even for small errors. However, this property introduces training instability, particularly during the crucial fine-tuning phase. As the model's predictions approach the ground truth, the persistently large gradient steps can cause the parameters to oscillate around the optima, preventing smooth and precise convergence and often degrading the final accuracy.

**The Dilemma: The Need for an Adaptive Signal.** The analysis of MSE and MAE reveals a fundamental dilemma: an ideal learning signal must be (1) strong and non-vanishing for small errors to learn fine-grained details (like MAE), yet (2) scaled by the error magnitude to ensure stable convergence (like MSE). No fixed loss function can satisfy both competing criteria. This necessitates a more sophisticated mechanism capable of generating an *adaptive* learning signal—one that is strong when it needs to correct meaningful details but gentle when it needs to stabilize. This motivates the use of a reinforcement learning framework, where the **advantage function** $A(s, a)$ can provide precisely such a dynamic signal, rewarding effective corrections strongly while ignoring marginal ones, thus resolving the gradient dilemma.

## 2.3 The Statistical View: The Bias-Variance Challenge in Correction

From a statistical perspective, the problem can be reframed as a bias-variance trade-off. Correcting the flaws of a base model requires reducing its bias without excessively increasing the variance of the final prediction.

**The High-Bias Nature of the Base Predictor and High-Variance Risk of Correction.** A model $f_\theta$ trained with MSE is inherently a **high-bias estimator** with respect to high-fidelity signals. It systematically underestimates both high-frequency dynamics and the magnitude of extreme events. Formally, its bias for these components, $\text{Bias}(\hat{y}) = \mathbb{E}[\hat{y}] - y$, is consistently large. To correct this, one could introduce a corrector module that learns a correction term $a$, yielding $\hat{y}_{corr} = \hat{y} + a$. The goal is for $a$ to be an unbiased estimator of the model's residual, i.e., $\mathbb{E}[a] \to -\text{Bias}(\hat{y})$. However, a powerful, unconstrained corrector can easily overfit to noise, introducing a large variance term, $\text{Var}(a)$. This high variance can overwhelm the gains from bias reduction, as the total error is a function of both: $\text{MSE}(\hat{y}_{corr}) = \text{Bias}(\hat{y}_{corr})^2 + \text{Var}(\hat{y}_{corr})$. A naive, end-to-end joint training of a corrector often falls into this trap.

**The Need for a Variance-Controlled Corrector.** This analysis reveals that a successful solution must be a **variance-controlled** corrector. It must be powerful enough to reduce the substantial biases of the base model while being rigorously constrained to ensure that the correction itself has low variance. This principle directly motivates the core components of SRRF. First, to constrain the search space of possible corrections, we need a strong contextual prior. **RAG** provides this by grounding the correction in plausible historical exemplars. Second, to ensure the learned correction policy is stable, we need explicit variance-reduction mechanisms. This is achieved through the integrated design of our **RL** agent, which employs techniques like temporal pooling, L2 regularization on the action magnitude, and the use of an advantage estimator to ensure that it learns an effective, yet stable and reliable, correction policy.

## 3 METHODOLOGY

As motivated in Section 2, to address the gradient dilemma of traditional losses and the bias-variance challenge in forecast correction, we propose the Self-adaptive Retrieval-augmented Reinforcement learning (SRRF) framework. This section delineates the proposed methodology, which integrates a retrieval-augmented framework with an RL-based self-reflection and self-correction mechanism designed to systematically reduce bias while controlling variance. The method is structured into three primary phases: retrieval of similar historical sequences, fusion of model predictions with retrieved references, and correction via an RL policy network. We begin by presenting the overall architecture in Section 3.1, followed by the construction of the external database in Section 3.2, and conclude with the RL-based self-reflection and correction mechanism in Section 3.3.

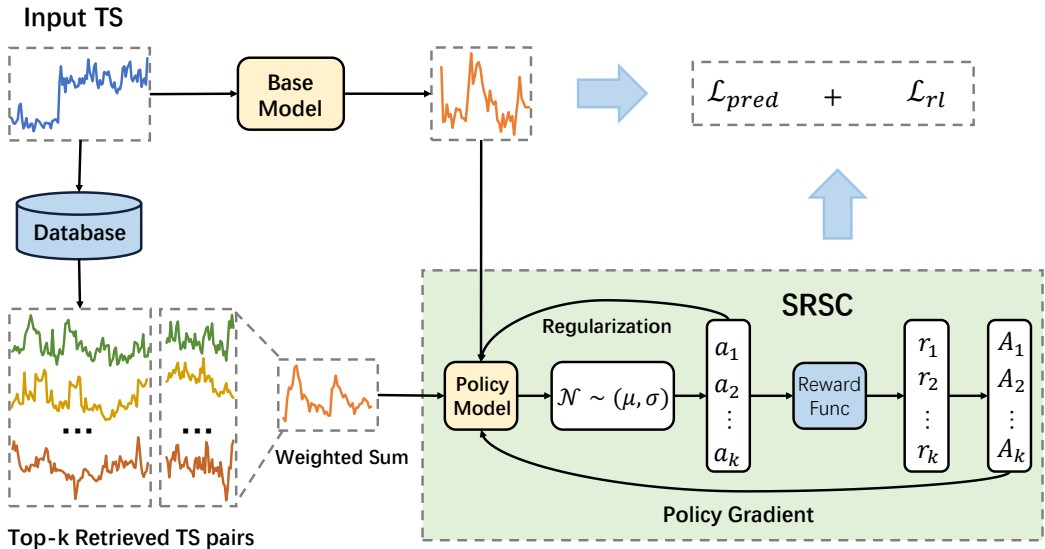

Figure 2: Overall architecture of the proposed SRRF framework. During training, an input series $\mathbf{x}$ is fed into a base model to produce an initial prediction $\hat{\mathbf{y}}$. In parallel, a reference $\mathbf{y}^{\text{ref}}$ is synthesized from retrieved historical exemplars. The state $s = [\hat{\mathbf{y}}; \mathbf{y}^{\text{ref}}]$ is used by an RL policy network to sample a correction term $a'$, leading to the final adjusted prediction $\mathbf{y}_{\text{adjusted}}$. The entire system is optimized with a combined loss $\mathcal{L}_{\text{total}}$, while the policy network is updated via policy gradients. At inference time, only the trained base model is used, incurring no extra overhead.

### 3.1 FRAMEWORK OVERVIEW

The proposed framework, illustrated in Figure 2, is a plug-and-play module built upon the `tslib` library, designed to enhance time series forecasting. It operates through three key phases:

- **Retrieval**: Retrieves the top-$k$ similar sequences from an external database $\mathcal{D}^R$ based on the input sequence, obtaining their corresponding ground-truth sequences as references.

- **Fusion**: Combines the model's initial prediction $\hat{y}$ with the retrieved reference $y^{\text{ref}}$ to compute a discrepancy signal, which is fed into the self-reflection and self-correction (SRSC) module.
- **Correction**: Evaluates the discrepancy between $\hat{y}$ and $y^{\text{ref}}$, samples a correction term using an RL policy network, and adjusts the prediction to produce the final output.

## 3.2 Building the External Database

To enable efficient retrieval, we construct an external database $\mathcal{D}^R$ from the training dataset. For each batch in the training set, input sequences of shape $[B, L, D]$ (where $B$ is the batch size, $L$ is the sequence length, and $D$ is the feature dimension) are flattened along the last two dimensions to shape $[B, F]$ (where $F = L \cdot D$) and stored in a FAISS index (Johnson et al., 2019) for fast similarity search using Euclidean distance. The corresponding ground-truth sequences of shape $[B, P, D]$ (where $P$ is the prediction length) are stored in a separate tensor to serve as references during retrieval. This process can be formally expressed as:

$$\mathcal{D}^R = \left\{ (\mathbf{x}_i, \mathbf{y}_i) \mid \mathbf{x}_i \in \mathbb{R}^F, \mathbf{y}_i \in \mathbb{R}^G, i = 1, 2, \ldots, N \right\}, \tag{1}$$

where $\mathbf{x}_i \in \mathbb{R}^F$ is the input sequence flattened along the sequence length and feature dimensions, $\mathbf{y}_i \in \mathbb{R}^G$ (with $G = P \cdot D$) is the corresponding ground-truth sequence, and $N$ is the total number of sequences in the training set. The FAISS index is constructed by adding all $\mathbf{x}_i$, while $\mathbf{y}_i$ are stored separately for retrieval. This approach ensures that $\mathcal{D}^R$ captures the full diversity of historical patterns, facilitating accurate retrieval of relevant sequences.

## 3.3 Self-Reflection and Self-Correction Based on Reinforcement Learning

To elevate the predictive performance, we introduce a sophisticated self-reflection and self-correction mechanism driven by a lightweight policy network. This mechanism dynamically analyzes the discrepancy between the model's initial prediction and a reference synthesized from retrieved sequences, subsequently generating a correction term to refine the prediction, thereby enhancing robustness and accuracy.

**Retrieval Mechanism** For a given input sequence $\mathbf{x} \in \mathbb{R}^F$, we retrieve the top-$k$ most similar sequences from $\mathcal{D}^R$, which computes the Euclidean distance $d(\mathbf{u}, \mathbf{v}) = \|\mathbf{u} - \mathbf{v}\|_2$. The ground-truth values of the retrieved sequences $\{\mathbf{y}_i\}_{i=1}^k$ are aggregated to form a reference $y^{\text{ref}}$ via a weighted average:

$$y^{\text{ref}} = \sum_{i=1}^{k} \alpha_i \mathbf{y}_i, \quad \alpha_i = \frac{\exp(-d_i)}{\sum_{j=1}^{k} \exp(-d_j)}, \tag{2}$$

where $d_i = \|\mathbf{x} - \mathbf{x}_i\|_2$ is the distance to the $i$-th retrieved sequence, $\mathbf{d} = \{d_i\}_{i=1}^k$, and $\epsilon$ is a small constant to prevent division by zero. This formulation ensures that sequences closer to the query contribute more significantly to the reference.

**Self-Reflection and Correction** The policy network observes the state signal $s = [y^{ref}; \hat{y}]$, capturing the discrepancy between the reference and the initial prediction $\hat{y}$. To map this high-dimensional state to specific correction parameters, we employ a lightweight Multi-Layer Perceptron (MLP) as the policy projector. Formally, the policy network $\pi_\phi$, parameterized by $\phi$, projects the state $s$ to the mean $\mu$ and the logarithm of the standard deviation $\log \sigma$:

$$[\mu; \log \sigma] = \text{MLP}_\phi(s) \tag{3}$$

where the output dimensions for both $\mu$ and $\sigma$ match the prediction shape $P \times D$. We output $\log \sigma$ to ensure numerical stability and apply a standard activation to guarantee positive variance. From these parameters, the correction term $a$ is sampled element-wise:

$$a \sim \mathcal{N}(\mu, \sigma^2). \tag{4}$$

To model the temporal dependencies among time steps, which are initially treated as independent by the policy network, we apply a temporal pooling operation to the correction term. This operation can be defined as:

$$a' = \text{Pool}(a; \kappa, \tau, \rho), \tag{5}$$

where $\text{Pool}(\cdot)$ denotes an average pooling function with kernel size $\kappa$, stride $\tau$, and padding $\rho$, ensuring that the correction term captures local temporal correlations, enhancing the coherence of the adjusted prediction.

**Justification for Temporal Pooling**   This pooling step is a crucial design choice for ensuring temporal coherence. The policy network initially predicts parameters for the correction term independently at each timestep. Without pooling, a direct sample from this distribution could result in a noisy correction signal where adjacent timesteps are uncorrelated, potentially introducing spurious spikes or "glitches" into the forecast. The temporal pooling operation acts as a smoothing filter on this raw signal. By averaging over a local window, it enforces that adjustments are correlated across neighboring timesteps, promoting a structured and plausible correction rather than erratic, point-wise noise. The final adjusted prediction is computed as:

$$y_{\text{adjusted}} = \hat{y} + a'. \tag{6}$$

The effectiveness of the correction is evaluated using reward functions based on the Mean Squared Error (MSE) and Mean Absolute Error (MAE):

$$r_{\text{MSE}} = \text{MSE}(\hat{y}, y_{\text{gt}}) - \text{MSE}(y_{\text{adjusted}}, y_{\text{gt}}), \quad r_{\text{MAE}} = \text{MAE}(\hat{y}, y_{\text{gt}}) - \text{MAE}(y_{\text{adjusted}}, y_{\text{gt}}), \tag{7}$$

where $y_{\text{gt}}$ is the true target. The total reward is defined as:

$$r = \begin{cases} 2, & \text{if} \quad r_{\text{MSE}} > 0 \quad and \quad r_{\text{MAE}} > 0, \\ 1, & \text{if} \quad r_{\text{MSE}} > 0 \quad or \quad r_{\text{MAE}} > 0, \\ 0, & \text{otherwise} \end{cases} \tag{8}$$

A positive reward indicates that the adjusted prediction is closer to the ground truth than the initial prediction. We opt for this discrete reward scheme as our early experiments revealed that directly using continuous reward values often led to high variance and numerical instability in policy gradient updates. The discrete signal provides a clearer and more stable learning signal for credit assignment.

**Optimization and Regularization**   The policy network is optimized using an RL objective inspired by Shao et al. (2024). For each sampled correction $a'_j$ (where $j = 1, 2, \ldots, N$, and $N$ is the number of samples), we compute the reward $r_j$ and normalize it to obtain the advantage:

$$A_j = \frac{r_j - \text{mean}(\mathbf{r})}{\text{std}(\mathbf{r}) + \epsilon}, \tag{9}$$

where $\mathbf{r} = \{r_j\}_{j=1}^N$, $\epsilon$ is a small constant for numerical stability. The RL loss is defined as:

$$\mathcal{L}_{\text{RL}} = -\mathbb{E}_{[\{a_j\}_{j=1}^N \sim \pi(a|s)]} \left[ \frac{1}{N} \sum_{j=1}^N \log \pi(a_j|s) \cdot A_j \right] + \lambda \cdot \frac{1}{N} \sum_{j=1}^N \|a'_j\|_2, \tag{10}$$

where $\pi(a_j|s)$ is the probability of action $a_j$ given state $s$, and $\lambda$ is the regularization coefficient to constrain the magnitude of corrections. The total loss combines the supervised MSE loss and the RL loss with weights:

$$\mathcal{L}_{\text{total}} = \gamma_1 \cdot MSE(\hat{y}, y_{gt}) + \gamma_2 \cdot \mathcal{L}_{\text{RL}}, \tag{11}$$

where $\gamma_1$ and $\gamma_2$ balance the contributions of the two loss terms. This integrated approach ensures that the model leverages historical patterns through retrieval and dynamically refines predictions using RL, achieving superior performance in time series forecasting tasks.

# 4 EXPERIMENTS

In this section, we conduct a comprehensive set of experiments to validate the effectiveness of the proposed SRRF framework. We first present the main forecasting results against sota baselines. We then provide critical ablation studies to justify our core design choices. Finally, we analyze key properties of SRRF, including its impact on spectral fidelity and its sensitivity to hyperparameters.

## 4.1 EXPERIMENTAL SETUP

**Datasets and Baselines.** Our evaluation is performed on five real-world multivariate time-series benchmarks: **ECL**, **ETT** (with four sub-datasets), **Exchange**, **Traffic**, and **Weather**. We compare SRRF against ten well-acknowledged forecasting models, encompassing Transformer-based, Linear-based, TCN-based, and LLM-based architectures.

**Evaluation Protocol.** We adhere to the standard data processing and evaluation protocols established by recent benchmark studies (Wu et al., 2022) to ensure a fair and direct comparison. The primary metrics are Mean Squared Error (MSE) and Mean Absolute Error (MAE). For reproducibility, detailed descriptions of the datasets, baseline implementations, and full experimental settings are provided in Appendix C.

## 4.2 MAIN RESULTS

We apply SRRF on top of iTransformer, a strong baseline, and compare its performance against the original model and other state-of-the-art methods. Table 1 summarizes the average MSE and MAE over all prediction horizons. The results demonstrate that SRRF consistently yields substantial performance gains, outperforming not only its base model but also other strong competitors across most datasets. This highlights the widespread nature of the spectral bias problem and the general effectiveness of SRRF as a solution.

Table 1: Forecasting results with prediction lengths $S \in \{96, 192, 336, 720\}$ and fixed lookback length $T = 96$. Results are averaged from all prediction lengths. *Avg* means further averaged by subsets. The result of **SRRF** is based on iTransformer (Liu et al., 2023). Full results are listed in Appendix D.

| Models | SRRF (Ours) | | TimeMixer (2024) | | iTransformer (2023) | | RLinear (2023) | | PatchTST (2023) | | ModernTCN (2024) | | FITS (2023) | | DLinear (2022) | | GPT4TS (2023) | |
|---|---|---|---|---|---|---|---|---|---|---|---|---|---|---|---|---|---|---|
| Metric | MSE | MAE | MSE | MAE | MSE | MAE | MSE | MAE | MSE | MAE | MSE | MAE | MSE | MAE | MSE | MAE | MSE | MAE |
| ETTh1 | **0.435** | **0.430** | 0.448 | 0.439 | 0.454 | 0.447 | 0.446 | 0.434 | 0.469 | 0.454 | 0.479 | 0.463 | 0.444 | 0.432 | 0.456 | 0.452 | 0.444 | 0.436 |
| ETTh2 | **0.374** | 0.399 | 0.389 | 0.411 | 0.383 | 0.407 | **0.374** | **0.398** | 0.387 | 0.407 | 0.413 | 0.430 | 0.381 | 0.402 | 0.558 | 0.515 | 0.379 | 0.406 |
| ETTm1 | **0.378** | **0.396** | 0.385 | 0.396 | 0.407 | 0.410 | 0.414 | 0.408 | 0.387 | 0.400 | 0.417 | 0.420 | 0.402 | 0.400 | 0.403 | 0.406 | 0.382 | 0.399 |
| ETTm2 | 0.277 | **0.322** | 0.276 | 0.324 | 0.288 | 0.332 | 0.286 | 0.327 | 0.280 | 0.326 | 0.321 | 0.360 | 0.285 | 0.325 | 0.350 | 0.400 | 0.280 | 0.325 |
| Weather | 0.256 | 0.276 | **0.243** | **0.273** | 0.258 | 0.278 | 0.272 | 0.290 | 0.258 | 0.280 | 0.246 | 0.274 | 0.250 | 0.278 | 0.265 | 0.316 | 0.260 | 0.280 |
| Exchange | **0.350** | **0.399** | 0.379 | 0.420 | 0.360 | 0.403 | 0.379 | 0.416 | 0.367 | 0.404 | 0.399 | 0.445 | 0.375 | 0.414 | 0.354 | 0.414 | 0.360 | 0.403 |
| ECL | **0.172** | **0.268** | 0.185 | 0.274 | 0.178 | 0.270 | 0.218 | 0.298 | 0.205 | 0.290 | 0.188 | 0.289 | 0.225 | 0.307 | 0.211 | 0.300 | 0.277 | 0.364 |
| Traffic | **0.421** | **0.282** | 0.514 | 0.311 | 0.428 | 0.282 | 0.626 | 0.383 | 0.481 | 0.304 | 0.568 | 0.359 | 0.633 | 0.385 | 0.624 | 0.383 | 0.550 | 0.339 |

## 4.3 ABLATION STUDIES

We conduct two critical ablation studies to validate the core design choices of the SRRF framework.

**Plug-and-Play Generalization.** To demonstrate the broad, model-agnostic effectiveness of SRRF, we integrate it into seven diverse, state-of-the-art base models. As shown in Table 2, SRRF consistently improves performance regardless of the underlying architecture. The framework achieves an average MSE reduction of 7.23% and MAE reduction of 5.89% across all tested scenarios, with particularly significant improvements observed in datasets exhibiting complex temporal dynamics, such as Weather and Traffic. These results confirm SRRF's versatility as a general-purpose training enhancement that can be readily integrated into a wide variety of forecasting models.

Table 2: Performance promotion obtained by our SRRF framework. We report the average performance and the relative MSE reduction (Promotion).

| Models | | TimeMixer (2024) | | iTransformer (2023) | | RLinear (2023) | | PatchTST (2023) | | ModernTCN (2024) | | FITS (2023) | | DLinear (2022) | | GPT4TS (2023) | |
|---|---|---|---|---|---|---|---|---|---|---|---|---|---|---|---|---|---|
| Metric | | MSE | MAE | MSE | MAE | MSE | MAE | MSE | MAE | MSE | MAE | MSE | MAE | MSE | MAE | MSE | MAE |
| ETT | Original | 0.374 | 0.393 | 0.383 | 0.399 | **0.380** | **0.390** | 0.380 | 0.397 | 0.407 | 0.418 | 0.378 | 0.390 | 0.441 | 0.443 | 0.379 | 0.399 |
| | +SRRF | **0.372** | **0.392** | **0.373** | **0.393** | 0.382 | 0.392 | **0.370** | **0.311** | **0.376** | **0.293** | **0.377** | **0.293** | **0.409** | **0.293** | **0.374** | 0.397 |
| | Promotion | 0.5% | 0.3% | 2.6% | 1.5% | -0.5% | -0.5% | 2.7% | 27.6% | 8.2% | 42.6% | 0.2% | 2.3% | 7.8% | 51.1% | 1.3% | 0.5% |
| ECL | Original | 0.185 | 0.274 | 0.178 | 0.270 | 0.218 | **0.298** | 0.204 | 0.290 | 0.188 | 0.289 | 0.225 | 0.307 | 0.211 | 0.300 | 0.277 | 0.364 |
| | +SRRF | **0.179** | **0.272** | **0.172** | **0.268** | **0.206** | 0.308 | **0.186** | **0.275** | **0.180** | **0.278** | **0.224** | 0.307 | **0.181** | **0.272** | **0.207** | **0.303** |
| | Promotion | 3.3% | 0.7% | 2.8% | 0.7% | 5.7% | -3.2% | 9.6% | 5.4% | 4.4% | 3.9% | 0.4% | 0% | 16.5% | 10.2% | 33.8% | 20.1% |
| Traffic | Original | 0.514 | 0.311 | 0.428 | 0.282 | 0.626 | 0.383 | 0.481 | 0.304 | 0.568 | 0.359 | 0.633 | 0.385 | 0.624 | 0.383 | 0.550 | 0.339 |
| | +SRRF | **0.507** | **0.300** | **0.421** | 0.282 | 0.626 | **0.378** | **0.478** | **0.298** | **0.531** | 0.359 | **0.623** | **0.382** | 0.624 | **0.366** | **0.506** | **0.327** |
| | Promotion | 1.3% | 3.6% | 1.6% | 0.0% | 0% | 1.3% | 0.6% | 2.0% | 6.9% | 0% | 1.6% | 0.7% | 0% | 3.8% | 8.6% | 3.6% |
| Weather | Original | 0.243 | 0.273 | 0.258 | 0.278 | 0.272 | 0.290 | 0.258 | 0.280 | 0.246 | 0.274 | 0.250 | 0.278 | 0.265 | 0.316 | 0.260 | 0.280 |
| | +SRRF | 0.243 | **0.272** | **0.256** | **0.276** | **0.260** | **0.285** | **0.254** | **0.277** | **0.243** | **0.273** | **0.247** | **0.276** | **0.262** | **0.296** | **0.259** | **0.279** |
| | Promotion | 0% | 0.3% | 0.7% | 0.7% | 4.6% | 1.7% | 1.5% | 1.1% | 1.2% | 0.3% | 1.2% | 0.7% | 1.1% | 6.7% | 0.3% | 0.3% |
| Exchange | Original | 0.379 | 0.420 | 0.360 | 0.403 | 0.379 | 0.416 | 0.367 | 0.404 | 0.399 | 0.445 | **0.375** | **0.414** | 0.354 | 0.414 | 0.360 | 0.403 |
| | +SRRF | **0.354** | **0.399** | **0.350** | **0.399** | **0.365** | **0.405** | **0.354** | **0.399** | **0.388** | **0.413** | 0.380 | 0.418 | **0.308** | **0.379** | **0.353** | **0.398** |
| | Promotion | 7.1% | 5.2% | 2.8% | 1.0% | 3.8% | 3.2% | 3.7% | 1.2% | 2.8% | 7.9% | -1.5% | -0.8% | 15.0% | 9.2% | 1.9% | 1.1% |

**Necessity of the Reinforcement Learning Framework.** A crucial question is whether our RL-based framework is necessary over simpler, end-to-end differentiable alternatives. We compare SRRF against two baselines: (1) a **Joint** baseline, where a correction module is trained jointly with the base model via a standard MSE loss, and (2) an **RL-only** baseline, where only the RL policy is trained to correct a frozen base model. As shown in Table 3, the Joint baseline suffers from training instability and catastrophic performance degradation on complex datasets, indicating a destructive interference between learning objectives. The RL-only baseline also fails, as its reward signal vanishes when the base model improves. In contrast, our **SRRF** framework, which decouples the training via RL, achieves robust and superior performance, providing strong empirical evidence for the necessity of our design.

Table 3: Ablation study on the necessity of the RL framework on the Weather and ECL datasets. The *Joint* baseline demonstrates instability, while our *SRRF* approach is robust and effective.

| Method | Weather | | ECL | |
|---|---|---|---|---|
| | MSE | MAE | MSE | MAE |
| iTransformer (Baseline) | 0.1736 | 0.2148 | 0.1483 | 0.2404 |
| Joint (Differentiable) | 0.1717 | 0.2131 | 1.2805 | 0.8851 |
| RL-only | 0.2126 | 0.2666 | 0.8745 | 0.7610 |
| **SRRF (Ours)** | **0.1694** | **0.2102** | **0.1461** | **0.2390** |

Table 4: Sensitivity analysis for the RL loss weight $\gamma_2$ (with $\gamma_1 = 1.0$) on iTransformer (Input-96, Predict-96). Best results for each dataset are in **bold**.

| $\gamma_2$ | Weather (MSE / MAE) | ECL (MSE / MAE) |
|---|---|---|
| 0 (Baseline) | 0.1736 / 0.2148 | 0.1483 / 0.2404 |
| 0.1 | 0.1737 / 0.2170 | 0.1485 / 0.2423 |
| **0.5** | **0.1694 / 0.2102** | 0.1462 / 0.2393 |
| **1.0** | 0.1740 / 0.2180 | **0.1461 / 0.2390** |
| 5.0 | 0.1737 / 0.2170 | 0.1587 / 0.2508 |

## 4.4 PERFORMANCE ANALYSIS

We further analyze key properties of SRRF, including its impact on spectral fidelity and its sensitivity to hyperparameters.

**Spectral Fidelity Analysis.** A core claim of our work is that SRRF mitigates the spectral bias inherent in MSE-trained models. To validate this claim quantitatively, we analyze the Frequency Root Mean Squared Error (FRMSE). The aggregate results in Figure 3 demonstrate that SRRF consistently reduces frequency-domain error across a wide range of models and datasets, offering robust evidence of enhanced spectral fidelity.

To deconstruct this quantitative improvement and provide qualitative insight, we present a case study of the iTransformer model on the volatile Exchange dataset in Figure 5. The time-domain plot (left) vividly illustrates how SRRF corrects the overly smooth predictions of the base model, faithfully capturing the sharp, transient fluctuations of the ground truth. This visual enhancement has a clear basis in the frequency domain (right): SRRF successfully restores the high-frequency spectral power that the base model erroneously suppresses, leading to a predicted spectrum that aligns far more closely with that of the ground truth. This representative example, supported by additional visualizations in Appendix D.3, provides compelling qualitative evidence for SRRF's ability to counteract spectral bias by capturing fine-grained temporal patterns.

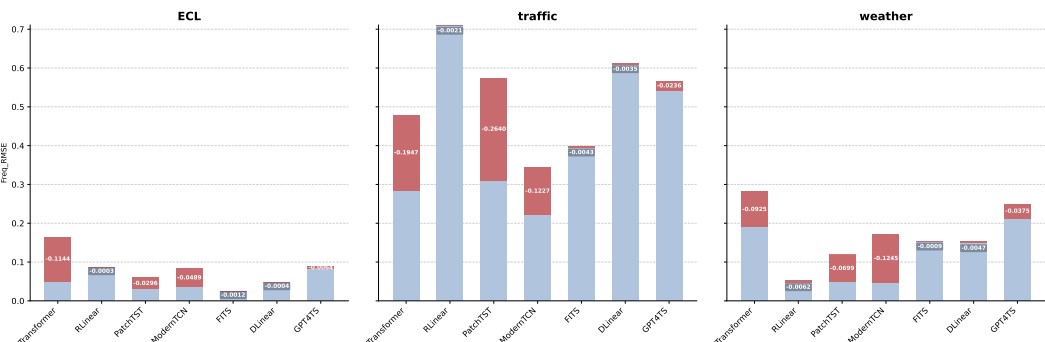

Figure 3: FRMSE before and after SRRF enhancement (Input-96, Predict-96). Blue bars correspond to base models; red bars correspond to SRRF-enhanced versions.

**Hyperparameter Sensitivity.** We analyze the sensitivity of SRRF to its two key hyperparameters: the retrieval exemplar count $k$ and the number of RL action samples $N_s$. Figure 4 summarizes the results across ETTh1, ECL, and Exchange. Increasing $N_s$ consistently improves performance when moving from 2 to 8 samples, but the gains taper off afterwards and mild overfitting appears at $N_s = 16$. This trend is observed across all datasets, suggesting that SRRF benefits from moderate stochastic exploration during RL training, whereas excessively large $N_s$ introduces noise or weakens the regularization effect. For the retrieval count $k$, using a single exemplar ($k = 1$) already provides a strong baseline. Aggregating a small neighborhood ($k \in [3, 5]$) typically yields the best performance by balancing contextual enrichment and noise suppression. Larger $k$ values may introduce redundant or irrelevant exemplars, which can dilute the useful signal and degrade accuracy.

We further investigate the sensitivity to the RL auxiliary loss weight $\gamma_2$, which governs the trade-off between the supervised base signal and the reinforcement learning correction signal. As detailed in Table 4, the performance remains robust across a broad interval $\gamma_2 \in [0.5, 1.0]$. Within this range, the RL signal provides sufficient gradient guidance to refine high-frequency details without overwhelming the base model's learning trajectory. In contrast, setting $\gamma_2$ too low (e.g., 0.1) results in negligible correction, rendering the RL component ineffective. Conversely, an excessively high weight (e.g., 5.0) can destabilize the training process, as the high-variance RL gradients may interfere with the convergence of the supervised objective. This bell-shaped response confirms that $\gamma_2$ acts as an effective regularizer, offering a stable sweet spot for practical deployment.

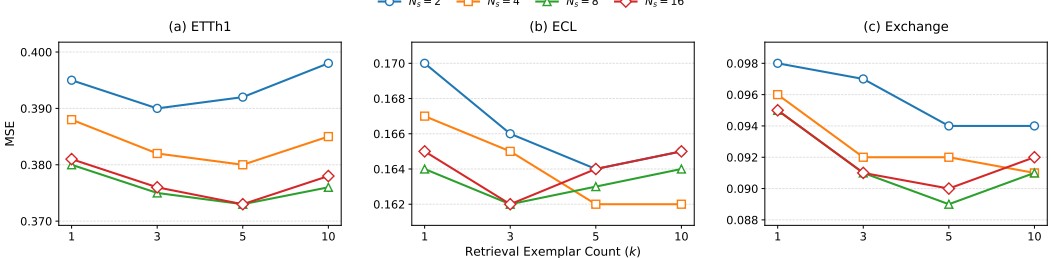

Figure 4: Impact of hyperparameters $k$ and $N_s$ on forecasting performance.

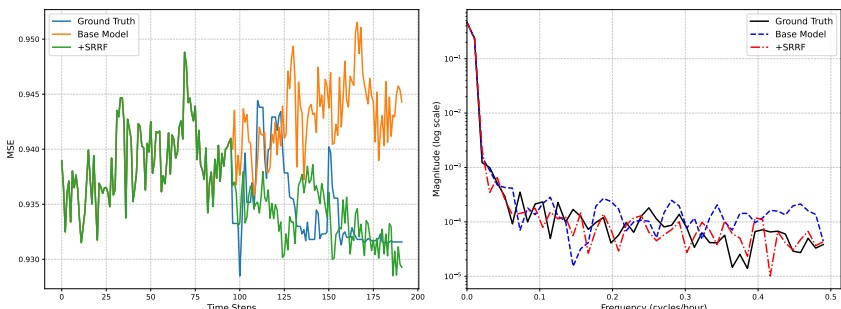

Figure 5: Time-domain and frequency-domain prediction results of iTransformer on the Exchange dataset.

## 5 DISCUSSION

We introduce SRRF, a plug-and-play training enhancement that combines RAG with RL to mitigate the spectral bias of MSE-optimized time series forecasting models. Our experiments show that SRRF consistently improves accuracy across diverse state-of-the-art architectures, with pronounced benefits for volatile and long-horizon forecasting. Nevertheless, SRRF incurs extra training costs due to retrieval and RL sampling, and its Euclidean distance-based retrieval may overlook complex similarities. Future directions include improving training efficiency, exploring more expressive retrieval mechanisms, and extending SRRF to tasks such as anomaly detection, classification, and imputation.

### ETHICS STATEMENT

This research adheres to the ICLR Code of Ethics. Our work is foundational in nature, focusing on improving the technical accuracy and spectral fidelity of time series forecasting models. The potential positive societal impacts include more efficient resource management in sectors like energy and transportation, and improved modeling in climate science. We used only publicly available, anonymized benchmark datasets that contain no personally identifiable or sensitive information. While any forecasting technology could potentially be misused, our work does not introduce any new or specific vulnerabilities. The proposed SRRF framework is a general-purpose training enhancement and is not designed for any specific high-risk application. We foresee no direct negative societal impacts stemming from this research.

### REPRODUCIBILITY STATEMENT

To ensure the reproducibility of our results, we have made the following resources available. **Source Code:** The complete source code for the SRRF framework, along with scripts to reproduce all experiments presented in this paper, is provided in the supplementary materials. The code is available at the following URL: https://anonymous.4open.science/r/ACAC-9999/README.md. **Theoretical Claims:** The theoretical arguments presented in Section 2 are supported by detailed, step-by-step mathematical proofs in Appendix B, covering both the gradient dilemma and the bias-variance framework. **Datasets:** All datasets used in our empirical evaluation are publicly available benchmarks. We cite their original sources and provide a detailed description in Appendix C.1. Our data processing follows the standard protocol established by prior work, also detailed in the appendix. **Experimental Details:** All hyperparameters and implementation details for our experiments, including the configurations for all baseline models, are documented in Appendix C.2.

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

## LLM Usage Statement

In the preparation of this manuscript, we utilized a large language model (LLM) as a general-purpose writing assistant. The primary use of the LLM was for improving the clarity, grammar, and phrasing of the text. The LLM did not contribute to the core research ideation, the design of the SRRF framework, the experimental methodology, or the analysis of the results. All theoretical claims and empirical findings presented in this paper are the original work of the authors, who take full responsibility for the content of this submission.

## A  Related Work

### A.1  Existing Time Series Models

The success of deep learning in time series forecasting has given rise to a variety of architectures, each providing different inductive biases. Traditional statistical models such as Autoregressive Integrated Moving Average (ARIMA) (Shumway et al., 2017) and Exponential Smoothing (Hyndman & Athanasopoulos, 2018) have been widely used due to their simplicity and interpretability. However, these models often struggle with complex, non-linear, and high-dimensional data commonly found in modern applications.

The success of deep learning in time series forecasting has given rise to a variety of architectures, each providing different inductive biases. Recurrent Neural Networks (RNNs) based models, such as particularly Long Short-Term Memory (LSTM) networks (Hochreiter & Schmidhuber, 1997), have been successful in capturing temporal dependencies. Convolutional Neural Networks (CNNs) based models have also been adapted for time series forecasting by leveraging their ability to extract local patterns (donghao & wang xue, 2024). Meanwhile, transformer-based (Vaswani et al., 2017) models have gained popularity due to their effectiveness in handling long-range dependencies through attention mechanisms. Notable examples include the Informer (Zhou et al., 2021), which introduces a probabilistic sparse attention mechanism to efficiently handle long sequences, and the Autoformer (Wu et al., 2021), which incorporates a decomposition architecture to separate trend and seasonal components, and FedFormer (Zhou et al., 2022) combined Fourier transforms with attention to capture periodic features. The iTransformer (Liu et al., 2023) further innovates by applying attention and feed-forward networks on inverted dimensions, focusing on variate tokens to capture multivariate correlations. More recently, researchers have explored simple MLP-based architectures coupled with clever preprocessing; for example, channel decomposition or patching techniques (Zeng et al., 2022) allow MLP models to achieve competitive accuracy with much lower complexity. Each of these architectures excels under certain conditions; however, they generally operate under a static training paradigm and can struggle when patterns shift or when encountering novel behaviors not seen in training data.

### A.2  Retrieval-Augmented Generation for Time Series Forecasting

Retrieval-Augmented Generation (RAG)(Lewis et al., 2020) is a technique originally from natural language processing that combines retrieval of relevant information with generative models to improve performance. In the context of time series forecasting, RAG involves retrieving similar historical sequences to inform the forecast of future values.

Recent studies have explored the application of RAG to time series forecasting. For instance, (Tire et al., 2024) introduced Retrieval Augmented Forecasting (RAF), which integrates retrieval mechanisms into time series foundation models to enhance zero-shot forecasting performance. Their framework demonstrates improved accuracy across diverse time series domains by leveraging related time series examples.

Another approach by (Ravuru et al., 2024) proposes an agentic RAG framework for time series analysis, where a multi-agent system retrieves relevant prompts from a shared repository to improve predictions. This hierarchical architecture allows for specialized sub-agents to handle specific tasks, leading to better performance in various time series applications.

These works highlight the potential of RAG in enhancing time series forecasting by providing additional context and reducing the reliance on large training datasets.

## A.3 Reinforcement Learning for Time Series Forecasting

Reinforcement Learning (RL) has been applied to time series forecasting in various ways, including model selection, parameter tuning, and direct forecasting. In RL, an agent learns to make decisions by interacting with an environment to maximize cumulative rewards.

For time series forecasting, RL can be used to dynamically select or combine models based on their performance. (Fu et al., 2022) proposed an RL-based model combination framework that learns to assign weights to base models in an ensemble, adapting to non-stationary time series data.

Additionally, RL can be employed to train forecasting models directly. (Kuremoto et al., 2014) introduced a deep belief net (DBN) trained with a reinforcement learning algorithm called stochastic gradient ascent (SGA), demonstrating improved robustness and forecasting accuracy.

In our work, we utilize RL to adjust the predictions made by our forecasting model based on retrieved similar sequences, optimizing the adjustment policy to minimize forecasting errors. This approach is inspired by the need to refine forecasts using historical patterns and is, to the best of our knowledge, a novel application of RL in this context.

## B    Detailed Theoretical Proofs

This appendix provides the formal theoretical underpinnings for the SRRF framework. We begin by mathematically establishing the core problem: the insufficiency of standard loss functions like MSE and MAE for high-fidelity time series forecasting. We then prove how SRRF serves as a principled solution to the challenges identified.

### B.1    Analysis of the Gradient Dilemma

Our core argument begins with a formal analysis of why standard gradient-based optimization with conventional losses fails to capture high-frequency details.

**Spectral Analysis of the MSE Gradient.**    The Mean Squared Error (MSE) loss is defined as $\mathcal{L}_{MSE}(\theta) = \frac{1}{T} \sum_{t=0}^{T-1} (y_t - \hat{y}_t(\theta))^2$. Using the discrete Fourier transform (DFT), where $Y(\omega) = \mathcal{F}(y_t)$ and $\hat{Y}(\omega; \theta) = \mathcal{F}(\hat{y}_t(\theta))$, Parseval's theorem allows us to express the loss in the frequency domain:

$$\mathcal{L}_{MSE}(\theta) = \frac{1}{T^2} \sum_{\omega=0}^{T-1} |Y(\omega) - \hat{Y}(\omega; \theta)|^2 \tag{12}$$

To analyze the learning dynamics, we can conceptualize the model's prediction for a specific frequency $\omega$ as being controlled by a set of parameters $\theta_\omega$. The gradient of the loss with respect to these parameters is therefore:

$$\frac{\partial \mathcal{L}_{MSE}}{\partial \theta_\omega} \propto -\left( Y(\omega) - \hat{Y}(\omega; \theta) \right) \tag{13}$$

During gradient descent, the magnitude of the parameter update for frequency $\omega$ is $|\Delta\theta_\omega| \propto |Y(\omega) - \hat{Y}(\omega; \theta)|$. In the early stages of training, or for components the model has not yet learned, $\hat{Y}(\omega; \theta) \approx 0$. Thus, the update magnitude is approximately proportional to the magnitude of the true signal's frequency component:

$$|\Delta\theta_\omega| \propto |Y(\omega)| \tag{14}$$

This proportionality is the mathematical origin of **spectral bias**. For real-world time series where low-frequency components have vastly larger energy (i.e., larger $|Y(\omega)|$), their corresponding parameters receive proportionally larger gradient updates. The learning signal for low-energy, high-frequency components is thus comparatively minuscule, causing the model to systematically underfit them.

**The Optimization Challenge of MAE.**    A natural alternative to address the vanishing gradient for small errors is the Mean Absolute Error (MAE) loss, $\mathcal{L}_{MAE} = \frac{1}{T} \sum_{t=0}^{T-1} |y_t - \hat{y}_t|$. Its gradient with respect to a prediction $\hat{y}_t$ is:

$$\frac{\partial \mathcal{L}_{MAE}}{\partial \hat{y}_t} = -\text{sgn}(y_t - \hat{y}_t) \tag{15}$$

The magnitude of the gradient is constant, $|\frac{\partial \mathcal{L}_{MAE}}{\partial \hat{y}_t}| = 1$ (for $y_t \neq \hat{y}_t$). While this solves the vanishing gradient problem, it introduces a critical optimization challenge. During the fine-tuning stage of training, when the prediction $\hat{y}_t$ is already very close to the true value $y_t$, the MAE loss continues to provide a large, constant-magnitude update signal. This can cause the optimization process to repeatedly "overshoot" the minimum, leading to oscillations and preventing smooth, precise convergence.

**The Dilemma.** The analysis of MSE and MAE reveals a fundamental **gradient dilemma**: an ideal loss function should provide a learning signal that is (a) strong and non-vanishing for small errors to counteract spectral bias, but also (b) scaled by the magnitude of the error to ensure stable convergence. Neither MSE nor MAE satisfies both criteria. This necessitates a more advanced mechanism that can generate an adaptive learning signal.

## B.2 SRRF as a Principled, Variance-Controlled Solution

Having established the problem, we now prove how SRRF provides a systematic solution from two complementary perspectives.

**The Advantage Function as an Adaptive Learning Signal.** The Reinforcement Learning (RL) agent in SRRF is updated via a policy gradient method. The gradient of the objective $J(\phi) = \mathbb{E}_{a \sim \pi_\phi}[r(a)]$ is estimated using an advantage function $A(s, a)$:

$$\nabla_\phi J(\phi) = \mathbb{E}_{a \sim \pi_\phi}[\nabla_\phi \log \pi_\phi(a|s) \cdot A(s, a)] \tag{16}$$

In our implementation, the advantage for a specific sampled action $a_j$ is empirically estimated and normalized:

$$A_j = \frac{r_j - \bar{r}}{\text{std}(\mathbf{r}) + \epsilon}, \quad \text{where} \quad \bar{r} = \frac{1}{N_s} \sum_{i=1}^{N_s} r(a_i) \tag{17}$$

The advantage function $A_j$ resolves the gradient dilemma. Our reward function $r$ is designed to be highly sensitive to high-frequency corrections (via its MAE component). When a corrective action $a_j$ is highly effective (i.e., successfully captures a high-frequency detail), it yields a high reward $r_j \gg \bar{r}$, resulting in a large positive advantage $A_j$. This provides a **strong, targeted update signal**. Conversely, when an action is only marginally effective, $r_j \approx \bar{r}$, resulting in $A_j \approx 0$. This provides a **vanishing update signal**, ensuring stability. The advantage function thus acts as an adaptive learning signal, providing strong updates only when necessary and ensuring stability otherwise.

**The Bias-Variance Framework for SRRF.** The second perspective demonstrates that SRRF is also a statistically robust framework for managing the bias-variance trade-off. The objective of the RL agent is to produce actions that counteract the bias of the base model: $\mathbb{E}[a] \to -\text{Bias}(\hat{y})$. The core challenge is to do so without excessively increasing the variance of the final prediction, $\text{Var}(\hat{y} + a)$. SRRF employs four mechanisms to provably control the variance of the correction term, $\text{Var}(a)$:

1. **RAG as a Conditional Prior.** By the law of total variance, conditioning on the retrieved reference $y^{\text{ref}}$ reduces variance:

   $$\text{Var}(a) = \mathbb{E}_{y^{\text{ref}}}[\text{Var}(a|s)] + \text{Var}_{y^{\text{ref}}}(\mathbb{E}[a|s]) \tag{18}$$

   A relevant $y^{\text{ref}}$ provides a strong prior, reducing the conditional variance $\text{Var}(a|s)$ and constraining the conditional mean $\mathbb{E}[a|s]$.

2. **Temporal Pooling.** For a sequence of independent corrections $\{a_t\}$ with average variance $\bar{\sigma}^2$, pooling over a window of size $\kappa$ yields a new sequence $\{a'_t\}$ with reduced variance:

   $$\text{Var}(a'_t) = \text{Var}\left(\frac{1}{\kappa} \sum_i a_i\right) = \frac{1}{\kappa^2} \sum_i \text{Var}(a_i) = \frac{\bar{\sigma}^2}{\kappa} \tag{19}$$

3. **Explicit L2 Regularization.** The RL loss includes the term $\mathcal{L}_{reg} = \lambda \cdot \mathbb{E}[\|a'\|_2]$. Minimizing this term penalizes the second moment $\mathbb{E}[(a')^2]$, which directly constrains the variance since $\text{Var}(a') = \mathbb{E}[(a')^2] - (\mathbb{E}[a'])^2$.

4. **Advantage-Based Gradient Estimation.** Using the advantage $A(s, a)$ instead of the raw reward $r(s, a)$ is a standard variance reduction technique for the policy gradient estimator itself, leading to more stable training and a lower-variance policy.

**Empirical Validation of Theoretical Necessity.**   Beyond the formal mathematical arguments, the necessity of our RL-based framework is further validated by empirically investigating simpler alternatives. As demonstrated in our ablation studies (Section 4.3), a fully differentiable **Joint** baseline suffers from catastrophic performance degradation on complex datasets. This empirical failure provides critical evidence that the learning objectives of the base predictor and the high-frequency corrector are often in conflict, leading to destructive interference under a single, monolithic loss. The empirical failure of this simpler alternative thus serves as a practical proof of the necessity for a decoupled training mechanism, which SRRF provides.

## C  IMPLEMENTATION DETAILS

### C.1  DATASET DESCRIPTIONS

To evaluate the performance of our proposed model, we conducted experiments on seven real-world datasets. These datasets include:

1. **ETT** (Zhou et al., 2021): This dataset contains 7 factors of electricity transformers from July 2016 to July 2018. It includes four subsets: ETTh1 and ETTh2 are recorded hourly, while ETTm1 and ETTm2 are recorded every 15 minutes.

2. **Exchange** (Wu et al., 2021): This dataset collects panel data of daily exchange rates from 8 countries from 1990 to 2016.

3. **Weather** (Wu et al., 2021): This dataset includes 21 meteorological factors collected every 10 minutes from the Weather Station of the Max Planck Biogeochemistry Institute in 2020.

4. **ECL** (Wu et al., 2021): This dataset records the hourly electricity consumption data of 321 clients.

5. **Traffic** (Wu et al., 2021): This dataset collects hourly road occupancy rates measured by 862 sensors on San Francisco Bay Area freeways from January 2015 to December 2016.

We followed the same data processing and train-validation-test set split protocol used in Times-Net (Wu et al., 2022), where the train, validation, and test datasets are strictly divided according to chronological order to ensure no data leakage issues. As for the forecasting settings, in the ETT, Weather, ECL, and Traffic datasets, we fixed the length of the lookback series as 96, and the prediction length varies in {96, 192, 336, 720}. The details of the datasets are provided in Table 5.

Table 5: Detailed dataset descriptions. *Dimension* denotes the variate number of each dataset. *Prediction Length* denotes the future time points to be predicted and four prediction settings are included in each dataset. *Dataset Size* denotes the total number of time points in (Train, Validation, Test) split respectively. *Frequency* denotes the sampling interval of time points.

| Dataset | Dimension | Prediction Length | Dataset Size | Frequency | Information |
|---|---|---|---|---|---|
| ETTh1, ETTh2 | 7 | {96, 192, 336, 720} | (8545, 2881, 2881) | Hourly | Electricity |
| ETTm1, ETTm2 | 7 | {96, 192, 336, 720} | (34465, 11521, 11521) | 15min | Electricity |
| Exchange | 8 | {96, 192, 336, 720} | (5120, 665, 1422) | Daily | Economy |
| Weather | 21 | {96, 192, 336, 720} | (36792, 5271, 10540) | 10min | Weather |
| ECL | 321 | {96, 192, 336, 720} | (18317, 2633, 5261) | Hourly | Electricity |
| Traffic | 862 | {96, 192, 336, 720} | (12185, 1757, 3509) | Hourly | Transportation |

## C.2 IMPLEMENTATION DETAILS

---

**Algorithm 1** SRRF Training Process (for one batch)

---

**Require:** Input lookback series $\mathbf{x} \in \mathbb{R}^{L \times D}$; Ground truth future series $\mathbf{y}_{gt} \in \mathbb{R}^{P \times D}$; Base forecasting model $M_{base}(\cdot; \theta_{base})$ with parameters $\theta_{base}$; Policy network $\pi_\phi(\cdot|\cdot; \phi)$ with parameters $\phi$; External retrieval database $\mathcal{D}^R$; Number of retrieved exemplars $k$; Number of RL action samples $N_s$

1: $\hat{\mathbf{y}} \leftarrow M_{base}(\mathbf{x}; \theta_{base})$            ▷ Obtain initial prediction from base model
                                                       ▷ **Phase 1: Retrieval (RAG)**
2: Retrieve $k$ most similar historical sequences $\{\mathbf{x}_j^{ret}, \mathbf{y}_j^{ret}\}_{j=1}^k$ from $\mathcal{D}^R$ based on $\mathbf{x}$
3: $\mathbf{y}^{\text{ref}} \leftarrow \text{WeightedAggregation}(\{\mathbf{y}_j^{ret}\}_{j=1}^k)$         ▷ $\mathbf{y}^{\text{ref}} \in \mathbb{R}^{P \times D}$
                       ▷ **Phase 2: RL-based Self-Reflection and Correction**
4: Define state for RL agent $s \leftarrow [\mathbf{y}^{\text{ref}}; \hat{\mathbf{y}}]$       ▷ ▷ Concatenated reference and prediction
5: Compute distribution params $[\mu, \log \sigma] \leftarrow \text{MLP}_\phi(s)$       ▷ ▷ Project state to action params
6: Initialize lists: $\mathbf{R}_{list} \leftarrow [], \mathbf{LogProbs}_{list} \leftarrow [], \mathbf{A}'_{list} \leftarrow []$
7: **for** $idx \leftarrow 1$ **to** $N_s$ **do**                            ▷ ▷ Sample $N_s$ actions
8:     Sample correction term $a_{idx} \sim \mathcal{N}(\mu, \sigma^2)$
9:     Apply temporal pooling $a'_{idx} \leftarrow \text{Pool}(a_{idx}; \kappa, \tau, \rho)$         ▷ ▷ $a'_{idx} \in \mathbb{R}^{P \times D}$
10:     Compute adjusted prediction $\mathbf{y}_{\text{adjusted},idx} \leftarrow \hat{\mathbf{y}} + a'_{idx}$
11:     $r_{idx} \leftarrow \mathcal{R}(\hat{\mathbf{y}}, \mathbf{y}_{\text{adjusted},idx}, \mathbf{y}_{gt})$                 ▷ ▷ Compute reward
12:     Append $r_{idx}$ to $\mathbf{R}_{list}$
13:     Append $\log \pi_\phi(a_{idx}|s)$ to $\mathbf{LogProbs}_{list}$
14:     Append $a'_{idx}$ to $\mathbf{A}'_{list}$
15: **end for**
16: Compute advantages $A_{idx} \leftarrow \text{Normalize}(\mathbf{R}_{list})[idx]$ for $idx = 1, \dots, N_s$
17: Compute RL loss
                    ▷ **Phase 3: Total Loss Computation and Parameter Optimization**
18: Compute base model loss $\mathcal{L}_{base} \leftarrow \text{MSE}(\hat{\mathbf{y}}, \mathbf{y}_{gt})$
19: Compute total loss $\mathcal{L}_{\text{total}} \leftarrow \gamma_1 \cdot \mathcal{L}_{base} + \gamma_2 \cdot \mathcal{L}_{\text{RL}}$
20: Update base model parameters $\theta_{base} \leftarrow \theta_{base} - \eta_{base} \nabla_{\theta_{base}} \mathcal{L}_{\text{total}}$
21: Update policy network parameters $\phi \leftarrow \phi - \eta_{policy} \nabla_\phi \mathcal{L}_{\text{RL}}$
22: **return** Updated $\theta_{base}, \phi$

---

All the experiments are implemented in PyTorch (Imambi et al., 2021) and conducted on a single NVIDIA A100 40GB GPU. We utilize ADAM (Kitaev et al., 2020) with an initial learning rate in $\{10^{-3}, 5 \times 10^{-4}, 10^{-4}\}$ and L2 loss for the model optimization. The batch size is uniformly set to $\{16, 32\}$ and the number of training epochs is fixed at 10. The number of retrievals is set to $\{3, 5\}$ and the number of samples is set to $\{4, 8\}$. All the compared baseline models that we reproduced are implemented based on the benchmark of TimesNet (Wu et al., 2022) Repository, which is fairly built on the configurations provided by each model's original paper or official code. We provide the pseudo-code of SRRF in Algorithm 1.

## C.3 COMPUTATIONAL OVERHEAD

To provide a clear picture of the trade-offs of our method, we conduct a comprehensive benchmark to quantify the overhead introduced by SRRF from two perspectives: training time and index construction.

**Training Time Overhead** We first measure the dynamic training cost. We test the iTransformer model on a broad set of datasets with a batch size of 128 on **two Tesla T4 GPUs** (one for FAISS index, one for model training). We measure the average training time per epoch (averaged over the first three epochs) for different numbers of RL action samples ($N_s$). The results are presented in Table 6. The results clearly show that the training time increases linearly with the number of samples $N_s$. This is expected, as each sample requires a forward pass through the policy network and reward calculation. It is crucial to reiterate, however, that this is a training-time overhead. As SRRF does not

alter the base model's architecture, the inference time and throughput remain identical to the baseline model.

Table 6: Training overhead analysis of SRRF on iTransformer (batch size 128, Tesla T4). Time is averaged per epoch.

| $N_s$ | ETTh1 (s/epoch) | Weather (s/epoch) | ECL (s/epoch) |
|---|---|---|---|
| 0 (Baseline) | 3.53 | 31.21 | 153.51 |
| 4 | 6.65 | 43.42 | 173.46 |
| 8 | 8.21 | 50.17 | 183.55 |
| 16 | 11.86 | 72.46 | 206.40 |

**Index Construction Overhead** Beyond the training throughput, we explicitly evaluate the initialization costs associated with the Retrieval-Augmented Generation (RAG) component. We conduct this evaluation on a single **NVIDIA RTX 3090 GPU**, using a batch size of 32 and FP32 precision for vector storage. Table 7 details the time required to build the FAISS index and its memory footprint across datasets with varying sizes and dimensions. The results demonstrate that the index construction is a highly efficient, one-time process, taking only seconds even for high-dimensional datasets like Traffic (862 features) or large-scale datasets like Weather. Regarding memory consumption, even in the most demanding cases (e.g., Traffic), the index occupies approximately 3.8 GB. This footprint is well within the capacity of modern GPUs or standard system RAM, confirming that the RAG component does not introduce prohibitive resource bottlenecks.

Table 7: Computational resource analysis for FAISS index construction. The evaluation uses a fixed input and prediction length of 96, a batch size of 32, and FP32 precision for vector storage on an RTX 3090 GPU.

| Dataset | Construction Time (s) | Memory Usage (MB) |
|---|---|---|
| ETTh1 | 1.15 | 21.66 |
| Weather | 2.58 | 282.21 |
| ECL | 8.92 | 2141.95 |
| Traffic | 11.73 | 3816.18 |

## D  FULL RESULTS

The full multivariate forecasting results are provided in the following section due to the space limitations of the main text.

### D.1  FULL PROMOTION RESULTS

We compare the performance of various strong baselines and their SRRF-enhanced counterparts across multiple benchmarks in Table 10. Consistent improvements can be observed for most models and datasets, demonstrating the general effectiveness of SRRF in enhancing long-term forecasting. These results suggest that spectrum-aware representations offer a promising direction for improving time series models across architectures and domains.

### D.2  VISUALIZATION OF FORECASTING RESULTS

To elucidate the impact of the SRRF framework on diverse base forecasting models (Liu et al., 2023; Nie et al., 2023; Li et al., 2023; donghao & wang xue, 2024; Xu et al., 2023; Zeng et al., 2022; Zhou et al., 2023), we present representative prediction cases from two datasets.

Figure 6 presents forecasting performance on the Traffic dataset, characterized by pronounced periodic variations. In this context, base models generally exhibit proficient performance in capturing the dominant, low-frequency cyclical components of the time series; as illustrated, the predictions

Table 8: Full results of the long-term forecasting task: Comparison of base models and models enhanced with SRRF. All prediction lengths are shown for each Model. Input sequence length is 96. Best performance for each model pair (Base vs. SRRF) is in **bold red**, second best in underlined blue.

| Model / Metric | TimeMixer (2024) | | | | iTransformer (2023) | | | | PatchTST (2023) | | | | DLinear (2022) | | | | RLinear (2023) | | | | ModernTCN (2024) | | | | FITS (2023) | | | | GPT4TS (2023) | | | |
|---|---|---|---|---|---|---|---|---|---|---|---|---|---|---|---|---|---|---|---|---|---|---|---|---|---|---|---|---|---|---|---|---|---|
| | Base | | +SRRF | | Base | | +SRRF | | Base | | +SRRF | | Base | | +SRRF | | Base | | +SRRF | | Base | | +SRRF | | Base | | +SRRF | | Base | | +SRRF | |
| | MSE | MAE | MSE | MAE | MSE | MAE | MSE | MAE | MSE | MAE | MSE | MAE | MSE | MAE | MSE | MAE | MSE | MAE | MSE | MAE | MSE | MAE | MSE | MAE | MSE | MAE | MSE | MAE | MSE | MAE | MSE | MAE |
| ETTh1 96 | 0.374 | 0.396 | 0.373 | 0.396 | 0.386 | 0.405 | 0.374 | 0.397 | 0.414 | 0.419 | 0.385 | 0.401 | 0.386 | 0.400 | 0.383 | 0.397 | 0.386 | 0.395 | 0.386 | 0.394 | 0.408 | 0.415 | 0.387 | 0.396 | 0.389 | 0.398 | 0.389 | 0.394 | 0.384 | 0.407 | 0.381 | 0.405 |
| ETTh1 192 | 0.442 | 0.432 | 0.441 | 0.432 | 0.441 | 0.436 | 0.427 | 0.426 | 0.460 | 0.445 | 0.430 | 0.432 | 0.437 | 0.432 | 0.433 | 0.426 | 0.437 | 0.424 | 0.435 | 0.422 | 0.467 | 0.447 | 0.437 | 0.424 | 0.441 | 0.424 | 0.440 | 0.424 | 0.431 | 0.428 | 0.429 | 0.423 |
| ETTh1 336 | 0.498 | 0.458 | 0.494 | 0.458 | 0.487 | 0.458 | 0.469 | 0.449 | 0.501 | 0.466 | 0.474 | 0.453 | 0.481 | 0.459 | 0.459 | 0.445 | 0.479 | 0.446 | 0.478 | 0.443 | 0.511 | 0.469 | 0.475 | 0.442 | 0.480 | 0.445 | 0.480 | 0.444 | 0.471 | 0.445 | 0.467 | 0.443 |
| ETTh1 720 | 0.478 | 0.472 | 0.493 | 0.476 | 0.503 | 0.491 | 0.474 | 0.473 | 0.500 | 0.488 | 0.481 | 0.481 | 0.519 | 0.516 | 0.515 | 0.514 | 0.481 | 0.470 | 0.478 | 0.467 | 0.531 | 0.523 | 0.484 | 0.464 | 0.466 | 0.462 | 0.466 | 0.459 | 0.501 | 0.475 | 0.473 | 0.462 |
| ETTh2 96 | 0.294 | 0.350 | 0.302 | 0.354 | 0.297 | 0.349 | 0.293 | 0.345 | 0.302 | 0.348 | 0.289 | 0.343 | 0.333 | 0.387 | 0.314 | 0.361 | 0.293 | 0.338 | 0.290 | 0.338 | 0.306 | 0.351 | 0.299 | 0.346 | 0.299 | 0.345 | 0.291 | 0.339 | 0.303 | 0.355 | 0.299 | 0.352 |
| ETTh2 192 | 0.376 | 0.396 | 0.365 | 0.391 | 0.380 | 0.400 | 0.374 | 0.395 | 0.388 | 0.400 | 0.375 | 0.394 | 0.477 | 0.476 | 0.428 | 0.440 | 0.374 | 0.390 | 0.377 | 0.393 | 0.430 | 0.440 | 0.382 | 0.396 | 0.383 | 0.395 | 0.380 | 0.393 | 0.381 | 0.405 | 0.380 | 0.404 |
| ETTh2 336 | 0.422 | 0.435 | 0.418 | 0.432 | 0.428 | 0.432 | 0.417 | 0.429 | 0.426 | 0.433 | 0.422 | 0.431 | 0.594 | 0.541 | 0.545 | 0.513 | 0.425 | 0.426 | 0.428 | 0.434 | 0.463 | 0.471 | 0.426 | 0.430 | 0.421 | 0.428 | 0.419 | 0.427 | 0.429 | 0.442 | 0.426 | 0.438 |
| ETTh2 720 | 0.464 | 0.464 | 0.428 | 0.442 | 0.427 | 0.445 | 0.416 | 0.437 | 0.431 | 0.446 | 0.423 | 0.443 | 0.831 | 0.657 | 0.682 | 0.586 | 0.420 | 0.440 | 0.439 | 0.450 | 0.453 | 0.457 | 0.411 | 0.432 | 0.421 | 0.438 | 0.419 | 0.437 | 0.433 | 0.453 | 0.427 | 0.449 |
| ETTm1 96 | 0.318 | 0.357 | 0.329 | 0.366 | 0.334 | 0.368 | 0.327 | 0.364 | 0.329 | 0.367 | 0.329 | 0.366 | 0.345 | 0.372 | 0.343 | 0.373 | 0.355 | 0.376 | 0.351 | 0.369 | 0.334 | 0.373 | 0.325 | 0.365 | 0.341 | 0.368 | 0.340 | 0.368 | 0.331 | 0.372 | 0.329 | 0.371 |
| ETTm1 192 | 0.366 | 0.386 | 0.369 | 0.385 | 0.377 | 0.391 | 0.373 | 0.383 | 0.367 | 0.385 | 0.356 | 0.380 | 0.380 | 0.389 | 0.376 | 0.390 | 0.391 | 0.392 | 0.388 | 0.386 | 0.390 | 0.406 | 0.370 | 0.393 | 0.380 | 0.386 | 0.379 | 0.385 | 0.373 | 0.394 | 0.372 | 0.394 |
| ETTm1 336 | 0.389 | 0.403 | 0.390 | 0.405 | 0.426 | 0.420 | 0.406 | 0.406 | 0.399 | 0.410 | 0.390 | 0.403 | 0.413 | 0.413 | 0.413 | 0.412 | 0.424 | 0.415 | 0.420 | 0.407 | 0.443 | 0.436 | 0.410 | 0.413 | 0.411 | 0.406 | 0.411 | 0.406 | 0.400 | 0.415 | 0.399 | 0.415 |
| ETTm1 720 | 0.465 | 0.440 | 0.455 | 0.442 | 0.491 | 0.459 | 0.474 | 0.445 | 0.454 | 0.439 | 0.444 | 0.435 | 0.474 | 0.453 | 0.472 | 0.451 | 0.487 | 0.450 | 0.480 | 0.440 | 0.502 | 0.466 | 0.490 | 0.453 | 0.476 | 0.441 | 0.476 | 0.440 | 0.474 | 0.464 | 0.462 | 0.463 |
| ETTm2 96 | 0.174 | 0.257 | 0.173 | 0.257 | 0.180 | 0.264 | 0.180 | 0.265 | 0.175 | 0.259 | 0.176 | 0.260 | 0.193 | 0.292 | 0.176 | 0.261 | 0.182 | 0.265 | 0.181 | 0.264 | 0.194 | 0.273 | 0.172 | 0.254 | 0.178 | 0.259 | 0.178 | 0.259 | 0.186 | 0.269 | 0.184 | 0.267 |
| ETTm2 192 | 0.236 | 0.297 | 0.238 | 0.299 | 0.250 | 0.309 | 0.249 | 0.310 | 0.241 | 0.302 | 0.248 | 0.307 | 0.284 | 0.362 | 0.261 | 0.315 | 0.246 | 0.304 | 0.260 | 0.312 | 0.292 | 0.349 | 0.239 | 0.297 | 0.248 | 0.299 | 0.247 | 0.305 | 0.253 | 0.315 | 0.249 | 0.310 |
| ETTm2 336 | 0.299 | 0.341 | 0.299 | 0.338 | 0.311 | 0.348 | 0.309 | 0.343 | 0.305 | 0.343 | 0.300 | 0.340 | 0.369 | 0.427 | 0.306 | 0.345 | 0.307 | 0.342 | 0.319 | 0.348 | 0.337 | 0.375 | 0.304 | 0.339 | 0.312 | 0.348 | 0.308 | 0.342 | 0.310 | 0.350 | 0.309 | 0.349 |
| ETTm2 720 | 0.394 | 0.402 | 0.389 | 0.395 | 0.412 | 0.407 | 0.411 | 0.405 | 0.402 | 0.400 | 0.394 | 0.395 | 0.554 | 0.522 | 0.424 | 0.406 | 0.407 | 0.398 | 0.435 | 0.413 | 0.464 | 0.443 | 0.412 | 0.400 | 0.404 | 0.394 | 0.407 | 0.397 | 0.409 | 0.409 | 0.408 | 0.406 |
| weather 96 | 0.160 | 0.207 | 0.162 | 0.209 | 0.174 | 0.214 | 0.169 | 0.207 | 0.177 | 0.218 | 0.175 | 0.216 | 0.196 | 0.255 | 0.172 | 0.221 | 0.192 | 0.232 | 0.172 | 0.222 | 0.156 | 0.204 | 0.156 | 0.205 | 0.166 | 0.214 | 0.166 | 0.214 | 0.177 | 0.217 | 0.176 | 0.216 |
| weather 192 | 0.207 | 0.251 | 0.206 | 0.250 | 0.221 | 0.254 | 0.220 | 0.253 | 0.225 | 0.259 | 0.225 | 0.253 | 0.237 | 0.296 | 0.236 | 0.268 | 0.240 | 0.271 | 0.236 | 0.268 | 0.208 | 0.253 | 0.206 | 0.250 | 0.215 | 0.256 | 0.214 | 0.256 | 0.225 | 0.258 | 0.223 | 0.258 |
| weather 336 | 0.262 | 0.290 | 0.261 | 0.289 | 0.278 | 0.296 | 0.278 | 0.296 | 0.278 | 0.297 | 0.269 | 0.295 | 0.283 | 0.335 | 0.283 | 0.312 | 0.292 | 0.307 | 0.278 | 0.301 | 0.272 | 0.297 | 0.266 | 0.293 | 0.271 | 0.296 | 0.266 | 0.292 | 0.281 | 0.298 | 0.282 | 0.299 |
| weather 720 | 0.343 | 0.344 | 0.340 | 0.340 | 0.358 | 0.347 | 0.357 | 0.349 | 0.354 | 0.348 | 0.349 | 0.345 | 0.345 | 0.381 | 0.358 | 0.359 | 0.364 | 0.353 | 0.355 | 0.351 | 0.351 | 0.342 | 0.347 | 0.345 | 0.350 | 0.347 | 0.345 | 0.343 | 0.357 | 0.348 | 0.356 | 0.344 |
| ECL 96 | 0.156 | 0.247 | 0.154 | 0.243 | 0.148 | 0.240 | 0.146 | 0.239 | 0.181 | 0.270 | 0.158 | 0.251 | 0.197 | 0.282 | 0.194 | 0.277 | 0.201 | 0.281 | 0.178 | 0.285 | 0.153 | 0.257 | 0.150 | 0.254 | 0.209 | 0.291 | 0.208 | 0.293 | 0.178 | 0.261 | 0.178 | 0.261 |
| ECL 192 | 0.169 | 0.260 | 0.168 | 0.259 | 0.162 | 0.253 | 0.162 | 0.253 | 0.188 | 0.274 | 0.169 | 0.262 | 0.196 | 0.285 | 0.156 | 0.248 | 0.201 | 0.283 | 0.197 | 0.301 | 0.179 | 0.281 | 0.165 | 0.266 | 0.208 | 0.293 | 0.207 | 0.293 | 0.293 | 0.389 | 0.200 | 0.303 |
| ECL 336 | 0.187 | 0.278 | 0.183 | 0.274 | 0.178 | 0.269 | 0.176 | 0.269 | 0.209 | 0.293 | 0.188 | 0.281 | 0.209 | 0.301 | 0.188 | 0.281 | 0.217 | 0.306 | 0.201 | 0.302 | 0.203 | 0.299 | 0.186 | 0.278 | 0.221 | 0.307 | 0.221 | 0.306 | 0.321 | 0.408 | 0.217 | 0.318 |
| ECL 720 | 0.227 | 0.312 | 0.210 | 0.301 | 0.225 | 0.317 | 0.207 | 0.309 | 0.246 | 0.324 | 0.232 | 0.316 | 0.245 | 0.333 | 0.203 | 0.299 | 0.257 | 0.331 | 0.239 | 0.334 | 0.222 | 0.318 | 0.220 | 0.315 | 0.262 | 0.337 | 0.262 | 0.337 | 0.318 | 0.400 | 0.235 | 0.333 |
| Exchange 96 | 0.108 | 0.232 | 0.083 | 0.201 | 0.086 | 0.206 | 0.084 | 0.204 | 0.088 | 0.205 | 0.084 | 0.202 | 0.088 | 0.218 | 0.077 | 0.196 | 0.093 | 0.217 | 0.082 | 0.200 | 0.092 | 0.218 | 0.083 | 0.202 | 0.093 | 0.216 | 0.092 | 0.216 | 0.086 | 0.203 | 0.085 | 0.202 |
| Exchange 192 | 0.187 | 0.311 | 0.175 | 0.297 | 0.177 | 0.299 | 0.176 | 0.299 | 0.176 | 0.315 | 0.156 | 0.290 | 0.184 | 0.307 | 0.179 | 0.299 | 0.180 | 0.308 | 0.176 | 0.297 | 0.187 | 0.309 | 0.186 | 0.309 | 0.179 | 0.300 | 0.172 | 0.295 | 0.179 | 0.300 | 0.172 | 0.295 |
| Exchange 336 | 0.359 | 0.434 | 0.324 | 0.412 | 0.331 | 0.417 | 0.326 | 0.415 | 0.301 | 0.397 | 0.300 | 0.397 | 0.313 | 0.427 | 0.261 | 0.385 | 0.351 | 0.432 | 0.332 | 0.417 | 0.352 | 0.427 | 0.340 | 0.418 | 0.335 | 0.420 | 0.334 | 0.420 | 0.333 | 0.419 | 0.322 | 0.411 |
| Exchange 720 | 0.862 | 0.701 | 0.833 | 0.686 | 0.847 | 0.691 | 0.813 | 0.678 | 0.901 | 0.714 | 0.854 | 0.699 | 0.839 | 0.695 | 0.737 | 0.645 | 0.886 | 0.714 | 0.866 | 0.702 | 0.972 | 0.828 | 0.953 | 0.734 | 0.885 | 0.713 | 0.911 | 0.726 | 0.841 | 0.688 | 0.833 | 0.685 |
| traffic 96 | 0.477 | 0.293 | 0.478 | 0.293 | 0.395 | 0.268 | 0.394 | 0.270 | 0.462 | 0.295 | 0.460 | 0.294 | 0.650 | 0.396 | 0.649 | 0.397 | 0.649 | 0.389 | 0.649 | 0.389 | 0.613 | 0.385 | 0.480 | 0.315 | 0.657 | 0.398 | 0.656 | 0.394 | 0.641 | 0.344 | 0.485 | 0.313 |
| traffic 192 | 0.504 | 0.313 | 0.505 | 0.312 | 0.417 | 0.276 | 0.407 | 0.275 | 0.466 | 0.296 | 0.466 | 0.278 | 0.598 | 0.370 | 0.597 | 0.370 | 0.605 | 0.366 | 0.602 | 0.367 | 0.501 | 0.335 | 0.494 | 0.330 | 0.609 | 0.374 | 0.605 | 0.371 | 0.489 | 0.310 | 0.482 | 0.308 |
| traffic 336 | 0.501 | 0.310 | 0.499 | 0.302 | 0.433 | 0.283 | 0.425 | 0.283 | 0.482 | 0.304 | 0.482 | 0.304 | 0.605 | 0.373 | 0.605 | 0.373 | 0.609 | 0.369 | 0.609 | 0.369 | 0.537 | 0.345 | 0.536 | 0.345 | 0.615 | 0.377 | 0.615 | 0.373 | 0.523 | 0.349 | 0.514 | 0.335 |
| traffic 720 | 0.575 | 0.328 | 0.546 | 0.293 | 0.467 | 0.302 | 0.457 | 0.300 | 0.514 | 0.322 | 0.507 | 0.317 | 0.645 | 0.394 | 0.646 | 0.325 | 0.647 | 0.387 | 0.647 | 0.387 | 0.622 | 0.370 | 0.614 | 0.367 | 0.653 | 0.393 | 0.652 | 0.391 | 0.548 | 0.353 | 0.544 | 0.352 |

from all seven base models manifest inherent periodicity, with principal distinctions observed in the rendition of peak magnitudes. The integration of SRRF, however, yields discernible improvements, particularly in the fidelity of peak forecasting. This enhancement is, nonetheless, contingent upon the capabilities of the underlying base model and notably more pronounced for Transformer-based architectures (e.g., iTransformer and PatchTST) compared to other model classes. This disparity can be primarily attributed to a key characteristic of Transformer architectures: the richer and more nuanced representations $\hat{y}$ learned by Transformers may provide a more conducive foundation upon which SRRF's reinforcement learning agent can develop a refined corrective policy specifically for sharp peak dynamics. In contrast, models with simpler architectures, such as linear variants (DLinear, RLinear), may possess insufficient capacity to inherently model the pronounced non-linearities characteristic of traffic peaks, thereby circumscribing the extent of refinement achievable by SRRF. Other architectures, like Temporal Convolutional Networks (TCNs, e.g., ModernTCN) or frequency-domain models (e.g., FITS), while effective for general pattern recognition, might exhibit inductive biases (e.g., fixed receptive fields in TCNs or specific spectral smoothing in FITS) that are less synergistic with SRRF's time-domain corrective actions for extreme peak events when compared to the dynamic and context-aware aggregation capabilities inherent in Transformers.

In contrast, Figure 7 showcases results on the Weather dataset, a series characterized by less distinct periodicity and a greater prevalence of stochastic, high-frequency fluctuations. On this dataset, the limitations of the base models in tracking such volatile dynamics are often more apparent. Their predictions frequently exhibit **significant deviations from the true underlying trend, an underestimation of the amplitude of high-frequency fluctuations, and a temporal lag in capturing abrupt changes**. The application of SRRF in this setting often results in more substantial improvements. The SRRF-enhanced models demonstrate a superior capability to adapt to these rapid, non-periodic variations, producing forecasts that align more closely with the ground truth's complex trajectory. As can be visually ascertained from the plots, SRRF exhibits a clear tendency to correct the initially deviated predictions of the base models towards the actual values. However, owing to the end-to-end joint training paradigm of SRRF with the base model, the ultimate performance enhancement remains intrinsically linked to the foundational capabilities of the base model; a stronger performing base model generally facilitates better performance from the SRRF-enhanced counterpart.

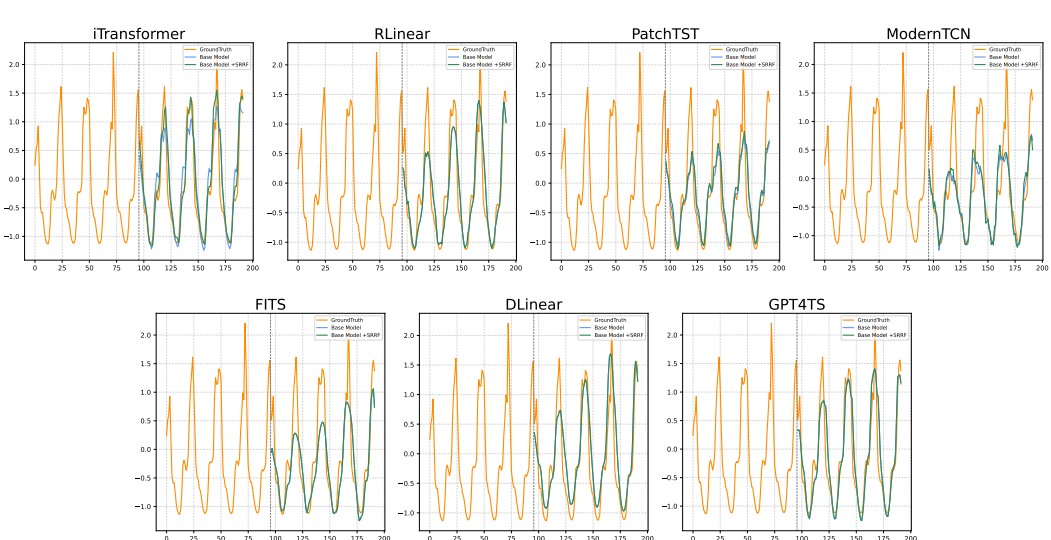

Figure 6: Forecasting results on the Traffic dataset (Input-96, Predict-96).

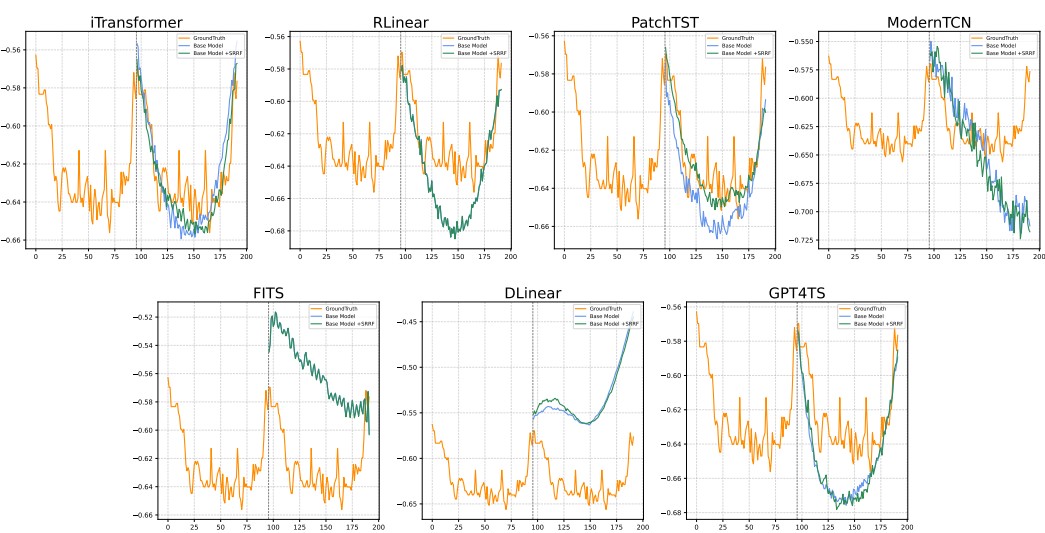

Figure 7: Forecasting results on the Weather dataset (Input-96, Predict-96).

Table 9: Overall FRMSE promotion and band-specific Prediction Error Energy Reduction ($L = 96, P = 96$). Frequency bands are normalized (0.5 Hz Nyquist). Improvements >50% are in **bold**.

| Dataset | Model | FRMSE | | FRMSE | Error Energy Reduction (%) in Frequency Band | | | | |
|---|---|---|---|---|---|---|---|---|---|
| | | Base | +SRRF | Promotion (%) | 0.0-0.1 Hz | 0.1-0.2 Hz | 0.2-0.3 Hz | 0.3-0.4 Hz | 0.4-0.5 Hz |
| ECL | iTransformer | 0.1649 | 0.0505 | **69.39** | -84.05 | **223.76** | 17.80 | 49.46 | 59.35 |
| | RLinear | 0.0850 | 0.0846 | 0.41 | -0.44 | 0.41 | 1.72 | 0.62 | 3.29 |
| | PatchTST | 0.0597 | 0.0301 | 49.58 | **243.30** | -25.40 | 0.53 | 18.85 | -0.88 |
| | ModernTCN | 0.0838 | 0.0348 | **58.44** | 217.41 | 9.10 | 33.57 | -13.83 | 63.69 |
| | FITS | 0.0243 | 0.0231 | 5.12 | -4.48 | -0.41 | 1.39 | 19.34 | 12.05 |
| | DLinear | 0.0462 | 0.0458 | 0.83 | 0.57 | 1.94 | 2.86 | -5.92 | 2.98 |
| | GPT4TS | 0.0885 | 0.0821 | 7.25 | -5.87 | -1.86 | 3.79 | 16.48 | 10.02 |
| Traffic | iTransformer | 0.4789 | 0.2842 | **40.65** | **132.02** | **168.42** | 40.03 | 15.20 | 11.24 |
| | RLinear | 0.7107 | 0.7085 | 0.30 | 1.35 | 2.09 | 1.52 | -1.34 | -0.22 |
| | PatchTST | 0.5744 | 0.3104 | **45.97** | **228.76** | **170.63** | **105.55** | **187.46** | -13.95 |
| | ModernTCN | 0.3440 | 0.2213 | **35.67** | -49.32 | 12.96 | -16.37 | -32.85 | **102.57** |
| | FITS | 0.3990 | 0.3947 | 1.07 | 1.65 | 2.16 | 2.66 | -1.39 | -1.88 |
| | DLinear | 0.6129 | 0.6094 | 0.57 | -0.94 | -0.50 | -1.41 | -0.31 | -13.37 |
| | GPT4TS | 0.5659 | 0.5422 | 4.18 | 11.36 | -1.60 | 3.96 | 13.39 | 6.60 |
| Weather | iTransformer | 0.2818 | 0.1893 | **32.82** | -52.32 | **90.43** | 15.10 | -35.81 | -8.91 |
| | RLinear | 0.0538 | 0.0476 | 11.52 | -18.13 | -10.23 | -16.44 | -14.25 | -2.82 |
| | PatchTST | 0.1185 | 0.0487 | **58.92** | -88.61 | **118.14** | -39.60 | **115.93** | -26.29 |
| | ModernTCN | 0.1714 | 0.0469 | **72.62** | -93.20 | -60.34 | **59.32** | -35.83 | 42.70 |
| | FITS | 0.1533 | 0.1524 | 0.57 | -1.01 | 0.49 | -4.60 | 10.06 | 5.70 |
| | DLinear | 0.1528 | 0.1481 | 3.07 | -5.47 | 11.44 | 9.62 | -23.29 | 19.74 |
| | GPT4TS | 0.2499 | 0.2124 | 15.02 | -26.24 | 12.64 | 7.07 | **120.10** | 32.04 |

## D.3 DETAILED ANALYSIS OF SRRF ON FREQUENCY DOMAIN RMSE

As discussed in Section 4.4 of the main text, the SRRF framework generally enhances spectral fidelity, measured by the overall Frequency Root Mean Squared Error (FRMSE). This section provides a more granular analysis by breaking down the prediction error energy into distinct frequency bands. Table 9 presents the percentage reduction of error energy, while Table 10 shows the raw error energy values before and after applying SRRF. This detailed view aims to substantiate our hypothesis that SRRF mitigates spectral bias by improving the representation of mid-to-high frequency information.

To quantitatively assess the SRRF framework's efficacy in enhancing the capture of diverse frequency components, we employ the Frequency Root Mean Squared Error (FRMSE). This metric evaluates the RMSE between the Discrete Fourier Transform (DFT) of the predicted series and that of the ground truth, offering insight into the spectral fidelity of the forecasts. Lower FRMSE values signify a more accurate reconstruction of the signal's amplitude spectrum. Figure 3 illustrates the overall FRMSE for the selected baseline models and their SRRF-enhanced versions across the ECL, Traffic, and Weather datasets, providing a global measure of spectral error. For a more detailed examination, Table 9 and Table 10 subsequently present an analysis of prediction error energy within specific frequency bands. This multi-faceted analysis aims to substantiate our hypothesis that SRRF mitigates spectral bias and improves the representation of high-frequency information.

The data in Table 9 reveals a complex pattern of spectral error redistribution, indicating that SRRF's impact is not uniform across all frequencies or models. First, examining the foundational 0.0-0.1 Hz band, which typically captures dominant trends and seasonal components, SRRF exhibits a notable "bidirectional" effect. On one hand, in certain instances, SRRF substantially enhances the model's ability to fit core low-frequency dynamics. For example, on the ECL dataset, PatchTST and ModernTCN achieved remarkable error energy reductions of 243.30% and 217.41%, respectively, indicating that SRRF helped them more accurately capture primary periodicities and trends. Similarly, on the Traffic dataset, iTransformer and PatchTST also achieved significant error energy reductions of 132.02% and 228.76%, respectively. These are highly positive indicators, suggesting SRRF's potential to improve predictions of the fundamental signal structure. On the other hand, we also observe instances where SRRF, conversely, increased the error energy in this lowest frequency band. For example, on the ECL dataset, iTransformer (-84.05%), on the Traffic dataset, ModernTCN (-49.32%), and on the Weather dataset, iTransformer (-52.32%), PatchTST (-88.61%), and ModernTCN

(-93.20%) all exhibited an increase in error energy (i.e., performance degradation) in the 0.0-0.1 Hz band. This phenomenon might imply that as SRRF strives to correct mid-to-high frequency details neglected by base models (due to MSE's spectral bias), its adjustment mechanisms might, in some cases, lead the model to make "sacrifices" in the fitting precision of the smoothest low-frequency components, or perhaps a "transfer" of error from higher to lower frequencies occurs.

Table 10: Prediction Error Energy (Base vs. +SRRF) and Improvement (%) across frequency bands ($L = 96, P = 96$). Frequency bands are normalized (0.5 Hz Nyquist). Positive Impr. % indicates error energy reduction by SRRF (improvement). Improvements >50% are in **bold**.

| Model | iTransformer (2023) | | | PatchTST (2023) | | | DLinear (2022) | | | RLinear (2023) | | | ModernTCN (2024) | | | FITS (2023) | | | GPT4TS (2023) | | |
|---|---|---|---|---|---|---|---|---|---|---|---|---|---|---|---|---|---|---|---|---|---|
| Metric | Base Energy | +SRRF Energy | Impr. % | Base Energy | +SRRF Energy | Impr. % | Base Energy | +SRRF Energy | Impr. % | Base Energy | +SRRF Energy | Impr. % | Base Energy | +SRRF Energy | Impr. % | Base Energy | +SRRF Energy | Impr. % | Base Energy | +SRRF Energy | Impr. % |
| **ECL** 0.0-0.1 | 14386.65 | 2294.07 | -84.1 | 537.93 | 1846.71 | **243.3** | 8791.90 | 8842.13 | 0.6 | 12680.77 | 12625.43 | -0.4 | 737.53 | 2340.97 | **217.4** | 2994.85 | 2860.62 | -4.5 | 21491.95 | 20231.42 | -5.9 |
| 0.1-0.2 | 12.06 | 39.04 | **223.8** | 13.12 | 9.79 | -25.4 | 6.20 | 6.32 | 1.9 | 62.71 | 62.97 | 0.4 | 52.05 | 56.78 | 9.1 | 16.78 | 16.71 | -0.4 | 45.70 | 44.85 | -1.9 |
| 0.2-0.3 | 7.41 | 8.72 | 17.8 | 7.85 | 7.89 | 0.5 | 1.39 | 1.43 | 2.9 | 17.00 | 17.29 | 1.7 | 10.39 | 13.88 | 33.6 | 1.37 | 1.39 | 1.4 | 8.09 | 8.40 | 3.8 |
| 0.3-0.4 | 3.63 | 5.42 | 49.5 | 4.32 | 5.14 | 18.8 | 0.10 | 0.09 | -5.9 | 19.65 | 19.77 | 0.6 | 7.16 | 6.17 | -13.8 | 0.00 | 0.00 | 19.3 | 1.56 | 1.81 | 16.5 |
| 0.4-0.5 | 1.71 | 2.72 | **59.4** | 1.51 | 1.49 | -0.9 | 0.02 | 0.03 | 3.0 | 19.49 | 20.13 | 3.3 | 3.83 | 6.27 | **63.7** | 0.00 | 0.00 | 12.1 | 1.52 | 1.68 | 10.0 |
| **Traffic** 0.0-0.1 | 134514.47 | 312101.78 | **132.0** | 43845.23 | 144147.03 | **228.8** | 226288.58 | 224163.80 | -0.9 | 38276.48 | 38792.35 | 1.3 | 80493.44 | 40793.63 | -49.3 | 112821.19 | 114685.78 | 1.7 | 139308.28 | 155137.06 | 11.4 |
| 0.1-0.2 | 13845.59 | 37164.70 | **168.4** | 10170.56 | 27524.41 | **170.6** | 14304.52 | 14233.28 | -0.5 | 4305.86 | 4395.75 | 2.1 | 1012.68 | 1143.90 | 13.0 | 12222.49 | 12027.21 | -1.6 | 5875.70 | 6108.24 | 4.0 |
| 0.2-0.3 | 3626.76 | 5078.51 | 40.0 | 1119.85 | 2301.87 | **105.6** | 748.04 | 737.51 | -1.4 | 394.70 | 400.70 | 1.5 | 356.04 | 297.76 | -16.4 | 845.08 | 867.56 | 2.7 | 5875.70 | 6108.24 | 4.0 |
| 0.3-0.4 | 182.00 | 209.66 | 15.2 | 124.57 | 358.10 | **187.5** | 29.14 | 29.04 | -0.3 | 67.93 | 67.02 | -1.3 | 468.26 | 314.45 | -32.8 | 0.01 | 0.01 | -1.4 | 274.63 | 311.39 | 13.4 |
| 0.4-0.5 | 16.20 | 18.02 | 11.2 | 70.76 | 60.89 | -13.9 | 0.57 | 0.49 | -13.4 | 93.10 | 92.89 | -0.2 | 141.31 | 286.25 | **102.6** | 0.00 | 0.00 | -1.9 | 176.85 | 188.53 | 6.6 |
| **weather** 0.0-0.1 | 39640.37 | 18902.46 | -52.3 | 6337.55 | 721.74 | -88.6 | 12809.06 | 12108.35 | -5.5 | 1972.25 | 1614.65 | -18.1 | 13356.35 | 908.40 | **-93.2** | 12925.99 | 12796.02 | -1.0 | 31359.90 | 23132.61 | -26.2 |
| 0.1-0.2 | 2.84 | 5.40 | **90.4** | 5.32 | 11.59 | **118.1** | 0.36 | 0.40 | 11.4 | 63.73 | 57.21 | -10.2 | 15.77 | 6.26 | -60.3 | 10.72 | 10.77 | 0.5 | 4.06 | 4.58 | 12.6 |
| 0.2-0.3 | 2.80 | 3.22 | 15.1 | 5.44 | 3.74 | -39.6 | 0.11 | 0.12 | 9.6 | 95.58 | 79.87 | -16.4 | 3.44 | 5.49 | **59.3** | 2.53 | 2.42 | -4.6 | 3.49 | 3.74 | 7.1 |
| 0.3-0.4 | 4.88 | 3.13 | -35.8 | 3.09 | 6.68 | **115.9** | 0.03 | 0.02 | -23.3 | 35.18 | 30.16 | -14.3 | 9.92 | 6.37 | -35.8 | 0.00 | 0.00 | 10.1 | 2.23 | 4.92 | **120.1** |
| 0.4-0.5 | 4.00 | 3.64 | -8.9 | 7.24 | 5.33 | -26.3 | 0.03 | 0.04 | 19.7 | 55.72 | 54.15 | -2.8 | 10.68 | 15.24 | 42.7 | 0.00 | 0.00 | 5.7 | 1.88 | 2.48 | 32.0 |

Next, we analyze the mid-to-high frequency bands (0.1-0.5 Hz). SRRF's performance in these bands is equally complex but more clearly reflects its potential to capture high-frequency details, which aligns with our primary goal of mitigating spectral bias. On the ECL dataset, in the 0.1-0.2 Hz band, iTransformer demonstrated an exceptionally large error energy reduction of 223.76%, a very significant improvement. ModernTCN also achieved a 63.69% improvement in the 0.4-0.5 Hz band on ECL. For the Traffic dataset, iTransformer reduced error energy in the 0.1-0.2 Hz band by 168.42%, while PatchTST achieved striking improvements across three consecutive bands: 170.63% (0.1-0.2 Hz), 105.55% (0.2-0.3 Hz), and 187.46% (0.3-0.4 Hz). ModernTCN also reduced error energy by 102.57% in the 0.4-0.5 Hz band on Traffic. In the volatile Weather dataset, iTransformer (0.1-0.2 Hz: 90.43%), PatchTST (0.1-0.2 Hz: 118.14%; 0.3-0.4 Hz: 115.93%), ModernTCN (0.2-0.3 Hz: 59.32%), and GPT4TS (0.3-0.4 Hz: 120.10%) all displayed significant error energy reductions in specific mid-to-high frequency bands. These instances provide strong evidence that SRRF can effectively enhance the model's ability to capture previously overlooked rapid changes and fine-grained patterns.

However, not all models exhibit performance improvements across all mid-to-high frequency bands. For instance, on the ECL dataset, PatchTST's error energy increased by 25.40% (i.e., a reduction percentage of -25.40%) in the 0.1-0.2 Hz band. On the Weather dataset, iTransformer also experienced increased error energy in the 0.3-0.4 Hz and 0.4-0.5 Hz bands. This inconsistency highlights the inherent difficulty of high-frequency component correction, which can be influenced by the quality of retrieved historical exemplars, the generalization capability of the RL policy, and the base model's own structural response characteristics to high-frequency signals.

In summary of these observations, although SRRF's performance in specific frequency bands (especially the lowest) can vary by model and dataset, sometimes even leading to increased error, its demonstrated ability to significantly reduce error energy in mid-to-high frequency bands supports SRRF's core value. This value lies in effectively mitigating the spectral bias from MSE optimization by enhancing learning in the mid-to-high frequency components, thereby promoting a more comprehensive spectral representation of time series.

