# OpenReview forum: "Self-adaptive Retrieval-Augmented Reinforcement Learning for Time Series Forecasting"
_ICLR.cc/2026/Conference — Submitted to ICLR 2026_

### Official Review · Reviewer_5Trs · 2025-10-29

**Soundness:** 2
**Presentation:** 2
**Contribution:** 3
**Rating:** 2
**Confidence:** 3

**Summary:**

The paper proposes Self-adaptive Retrieval-augmented Reinforcement Learning for Time Series Forecasting, a novel enhancement framework designed to address spectral bias in time series models trained with MSE. The framework integrates Retrieval-Augmented Generation to provide contextual grounding through historical exemplars and employs RL for adaptive correction of base model predictions. The paper demonstrates SRRF’s effectiveness through comprehensive experiments, showing significant improvements in forecasting performance across diverse datasets and models, especially in terms of spectral fidelity.

**Strengths:**

1. Innovative framework for spectral bias correction.
SRRF’s integration of RAG and RL to counteract spectral bias is novel and effectively addresses a long-standing challenge in time series forecasting. The clear focus on mitigating spectral bias enhances its applicability to volatile data where high-frequency dynamics are critical.

2. Plug-and-play design with no inference overhead.
The SRRF framework is a training-time enhancement that operates without changing the base model architecture. This approach ensures no added computational burden during inference, making SRRF a practical solution for enhancing existing models without requiring significant infrastructure changes.

**Weaknesses:**

1. Lack of clarity in RL correction mechanism.
The mathematical details of the RL correction process (Sec. 2.4) are not fully elaborated, leaving some uncertainty about the exact mechanism through which the RL agent refines predictions. While the qualitative results suggest success, the absence of a more detailed theoretical explanation or formal derivation of the RL correction step limits the depth of understanding regarding how SRRF resolves issues with traditional gradient-based optimization (Sec. 2).

2. Sensitivity to hyperparameters.
The performance of SRRF is sensitive to key hyperparameters, including the retrieval count (k) and the RL sample count, with noticeable degradation when too many exemplars are retrieved. While the paper discusses these sensitivities, it lacks a deeper exploration of how these parameters interact across different datasets, which could further help users tailor the framework to their needs.

3. Increased computational costs during training.
Although SRRF improves model accuracy, the additional computational cost from RAG and RL sampling may limit its scalability, particularly for large datasets or more complex base models (Sec. 5). The paper acknowledges these costs but does not provide a detailed breakdown of the time or memory overhead incurred by the retrieval and reinforcement learning steps during training.

4. While SRRF significantly enhances performance in high-frequency components, its impact on low-frequency components is sometimes inconsistent. In certain cases, SRRF leads to an increase in error energy in the lowest frequency band, suggesting that the framework may unintentionally sacrifice some predictive accuracy for smooth trends in favor of capturing finer details.

**Questions:**

1. A more explicit description or pseudocode would clarify how the reinforcement signal modifies predictions and how stability is ensured during joint optimization.

2. What is the empirical trade-off between retrieval depth and performance?
How sensitive are results to the number of retrieved exemplars and their selection strategy?

3. What portion of the total training time is attributable to retrieval and RL sampling? Would a lighter retrieval or policy model achieve similar gains? Including GPU hours or per-epoch runtime comparisons to baselines would strengthen the practicality argument.

---

> ### Author Response · Authors · 2025-11-26
>
> We sincerely thank you for recognizing the innovation of our work and the advantage of zero inference overhead. Your observation regarding the low-frequency components is particularly insightful. Based on your suggestions, we have comprehensively revised the paper, refining mechanism descriptions, pseudo-code, and overhead analysis. Below are our responses to your questions.
>
>
> **Q1: Lack of clarity in RL correction mechanism (Pseudocode or explicit description needed).**
>
> We have rewritten the description of the RL correction process in **Section 3.3** of the revised paper. Specifically, the policy network acts as a lightweight **MLP projector** that takes the concatenated state $s = [y^{ref}; \hat{y}]$ (comprising the reference and current prediction) and maps it directly to the Gaussian distribution parameters $[\mu; \log\sigma] = \text{MLP}(s)$. The correction term $a$ is then sampled from this distribution $\mathcal{N}(\mu, \sigma^2)$. To demonstrate this more intuitively, we have updated and refined Algorithm 1 in Appendix C.2, which clearly depicts the complete data flow from state construction and parallel sampling to reward calculation and gradient updates. Additionally, we provided detailed derivations on the Gradient Dilemma and Bias-Variance Trade-off in Section 2 and Appendix B, theoretically supporting how SRRF resolves traditional optimization issues.
>
>
> **Q2: Sensitivity to hyperparameters (Need exploration across different datasets).**
>
> To address this concern, we expanded **Figure 4** in the revised manuscript to include three subplots, displaying the impact of $k$ (retrieval count) and $N_s$ (sample count) on performance across three diverse datasets: ETTh1, ECL, and Exchange. The results show a consistent trend: performance gains typically stabilize when the sample count $N_s$ reaches 4 to 8. This consistency across datasets indicates that our default hyperparameter settings possess good robustness, sparing users from tedious tuning for each new dataset.
>
>
> **Q3: Increased computational costs during training (Need breakdown of time/memory overhead).**
>
> We added a dedicated section in Appendix C.3 to quantify these costs. **Table 7** shows that index construction is a one-time process and extremely fast (e.g., taking only 11.73 seconds on the largest Traffic dataset). **Table 6** presents the dynamic training overhead; taking iTransformer as an example, introducing SRRF increases the training time per epoch from 3.53 seconds to 6.65 seconds. While the training cost increases, it yields significant accuracy gains. Most importantly, SRRF introduces **zero additional overhead during the inference phase**, which is a massive advantage for practical deployment.
>
>
> **Regarding the inconsistent impact on low-frequency components (Weakness 4).**
>
> This is a very acute and profound observation. We discussed this in depth in the detailed spectral analysis in **Appendix D.3**. The standard MSE loss function inherently causes models to overfit high-energy low-frequency components (i.e., spectral bias), thereby ignoring high-frequency details. The core design intent of SRRF is to break this bias and force the model to focus on and restore lost high-frequency information. In some cases, this may indeed lead to a slight increase in fitting error in the low-frequency band (as shown in **Table 9**), but this is a necessary **trade-off** for achieving comprehensive spectral fidelity and significant overall prediction accuracy (lower MSE/MAE). From a practical application perspective, the ability to capture high-frequency dynamics like sudden changes and peaks is often more valuable than merely fitting smooth trends.

---

### Official Review · Reviewer_7A3k · 2025-10-30

**Soundness:** 3
**Presentation:** 3
**Contribution:** 3
**Rating:** 6
**Confidence:** 3

**Summary:**

This paper aims to address the problems of MSE loss function commonly used in the regression problem and thus proposes Retrieval-augmented Reinforcement learning for time series Forecasting (SRRF). The idea is to provide compensations via a policy network to the predicted outputs. The main model is trained via a joint loss function while the policy network is trained to minimize the RL loss via a policy gradient method.

**Strengths:**

1) the idea of RL for compensations of predicted output is novel;
2) the presentation of this paper is easy to follow;
3) Numerical results are convincing.

**Weaknesses:**

1) Code information should be placed in the abstract to gain better visibility;
2) Comparisons with those published in 2025 onward should be added;
3) The performance difference is not big such that the statistical tests are necessary;
4) Complexity analysis should be provided.

**Questions:**

1) Code information should be placed in the abstract to gain better visibility;
2) Comparisons with those published in 2025 onward should be added;
3) The performance difference is not big such that the statistical tests are necessary;
4) Complexity analysis should be provided.

---

> ### Author Response · Authors · 2025-11-26
>
> We sincerely thank you for your recognition of our work and the novelty of SRRF. Your suggestions have helped us further improve the presentation and validation of the paper. Based on your comments, we have made the following revisions.
>
>
> **W1: Code information should be placed in the abstract to gain better visibility.**
>
> We have fully adopted your suggestion. In the revised paper, we have explicitly added the link to the anonymous code repository to the last sentence of the Abstract to ensure immediate visibility.
>
>
> **W2: Comparisons with those published in 2025 onward should be added.**
>
> In the revised version (Table 1, Table 2 and Table 8), we have added **TimeMixer** as comparative baselines. The experimental results demonstrate that the SRRF-enhanced models continue to outperform these recent baselines.
>
>
> **W3: The performance difference is not big such that the statistical tests are necessary.**
>
> Regarding performance significance, we provide evidence from multiple perspectives to prove that the improvements are robust. First, we conducted robustness checks with different random seeds, showing that the fluctuation in MSE is negligible (< 0.001). In contrast, SRRF delivers performance gains that are often an order of magnitude larger. Second, beyond scalar metrics, our detailed spectral analysis in Appendix D quantitatively shows that SRRF significantly reduces the Frequency RMSE (FRMSE) in high-frequency bands, confirming that the model effectively restores fine-grained details lost by base models rather than merely fitting random noise.
>
> **W4: Complexity analysis should be provided.**
>
> We have added a dedicated "Computational Overhead" section in Appendix C.3 of the revised paper, providing **Table 6 and Table 7** to quantify the costs. The results show that index construction takes only seconds (e.g., 11.73s for the Traffic dataset), and while training time increases linearly, it remains manageable. Most importantly, SRRF introduces **zero additional overhead during the inference phase**.

---

### Official Review · Reviewer_Ah7n · 2025-10-31

**Soundness:** 3
**Presentation:** 3
**Contribution:** 2
**Rating:** 4
**Confidence:** 4

**Summary:**

To address the issue of overly smooth predictions caused by MSE-based training, the paper proposes a Self-adaptive Retrieval-augmented Reinforcement learning framework (SRRF) for time series forecasting. SRRF employs Retrieval-Augmented Generation to provide contextual grounding and reinforcement learning to correct initial forecasts. By integrating the SRRF module into various forecasting models, the approach achieves good prediction performance.

**Strengths:**

1. The paper proposes a plug-and-play that can be applied to all time series forecasting models.

2. The paper adopts reinforcement learning to learn a policy network that corrects the model’s initial forecasting results.

3. As a plug-and-play, SRRF achieves promising forecasting performance across different models.

**Weaknesses:**

W1. Method description is unclear:  1. The paper does not clearly explain how the policy network learns the mean (**μ**) and standard deviation (**σ**) from the reference prediction and the initial prediction. 2. It is also unclear whether the policy network generates the correction term (**α**) for each individual time step or for the entire time series sample.

W2. Experiment results: 1. The SRRF results in Table 1 are reported on top of the iTransformer, but they are inconsistent with the iTransformer+SRRF results shown in Table 2.   2. Some baseline results in Table 2 are significantly worse than those reported in the original papers — for example, PatchTST on the Traffic and Weather datasets shows noticeably lower prediction performance.

W3. Hyperparameter sensitivity experiment: The authors did not specify which dataset was used for this experiment.  Moreover, the RL sample count parameter has a major impact on prediction performance. The authors should conduct additional experiments across multiple datasets to verify whether the optimal choice of this parameter varies between datasets.

W4. Missing baselines: The authors should include more recent baselines, such as TimeMixer and TimeMixer++, for a more comprehensive comparison.

**Questions:**

See Weaknesses.

---

> ### Author Response · Authors · 2025-11-26
>
> We sincerely thank you for your detailed and professional review. We take your feedback very seriously and have comprehensively revised the paper, including correcting significant data typos, refining the method description, and adding new baselines. Below are our responses to your questions.
>
> **W1: Method description is unclear (How does the policy network learn $\mu, \sigma$? Is $a$ generated for each time step?)**
>
> We have elaborated on this mechanism in the revised Section 3.3 and Algorithm 1 in the Appendix. Specifically, we use a lightweight MLP as the policy projector, which receives the state $s = [y^{ref}; \hat{y}]$ and maps it to $\mu$ and $\log\sigma$ for the entire sequence length at once. While efficient, this initial sampling lacks explicit temporal dependency between steps. Therefore, a critical subsequent step is the application of Temporal Pooling (Eq. 5), which restores correlations across adjacent time steps through local smoothing, thereby preventing unrealistic high-frequency noise and ensuring the coherence of the correction.
>
>
> **W2: Experiment results (Inconsistencies between Table 1 & 2; PatchTST baseline results)**
>
> First and foremost, we offer our sincerest apologies. You were absolutely correct to flag the results for PatchTST+SRRF in Table 2; there was a significant typo in the previous version that erroneously exaggerated the performance gains. To rectify this serious error and resolve inconsistencies, we have **re-conducted the relevant experiments and performed a rigorous calibration of Table 1, Table 2, and Table 8 in Appendix D**. The corrected data shows that while the improvement for PatchTST+SRRF is not as massive as previously misreported, SRRF still delivers consistent and significant gains. We have also unified the statistical standards across tables to ensure data consistency. Thank you for helping us identify this oversight and ensuring the rigor of our work.
>
>
> **W3: Hyperparameter sensitivity experiment (Missing dataset spec; verify across multiple datasets)**
>
> In the revised manuscript, we have expanded Figure 4 to include three subplots, displaying hyperparameter sensitivity ($k$ and $N_s$) across three diverse datasets: ETTh1, ECL, and Exchange. The results indicate that despite differences in dataset characteristics, performance gains generally stabilize when the sample count $N_s$ reaches 4 to 8. This consistent trend across multiple datasets verifies the robustness of our default hyperparameter settings.
>
>
> **W4: Missing recent baselines (e.g., TimeMixer)**
>
> In the revised version, we have added TimeMixer to the comparative experiments in Table 1, Table 2 and Table 8. The latest results demonstrate that even against a strong, recent SOTA model like TimeMixer, models enhanced by SRRF continue to exhibit superior performance in most forecasting tasks, further validating the effectiveness of SRRF as a general-purpose enhancement framework.

---

### Official Review · Reviewer_ci19 · 2025-11-01

**Soundness:** 1
**Presentation:** 2
**Contribution:** 1
**Rating:** 0
**Confidence:** 3

**Summary:**

The authors propose to integrate RAG and RL into the learning process of a deep learning models for time series forecasting.

**Strengths:**

- original idea
- reasonably well written paper

**Weaknesses:**

- unclear integration of RAG and RL into model training
- no detailed analysis of additional training costs introduced by RAG and RL

**Questions:**

Given that RAG has high costs, how do you integrate RAG into the iterations used during the optimization process?
How can a subset of plausible historical examples be used to train the model, and how exactly are these examples selected?
How much extra training costs are caused by your RAG and RL sampling procedure?

---

> ### Author Response · Authors · 2025-11-26
>
> Thank you for your valuable feedback. We have refined the description of the policy network in the Methodology section (Section 3.3), and updated the pseudo-code (Algorithm 1) and overhead experiments (Table 7) in the Appendix to provide a more intuitive explanation. Below are our direct responses to your questions.
>
> **Q1: Given that RAG has high costs, how do you integrate RAG into the iterations used during the optimization process?**
>
> To address the cost issue, we construct a one-time vector index using the FAISS library prior to training and **store it on the GPU**, which allows for direct and efficient top-k retrieval during training. The revised Algorithm 1 (Appendix C.2) demonstrates the complete workflow: after each training batch $x$ is input, the top-k similar sequences are retrieved in real-time via the FAISS index and weighted aggregated into a reference signal $y^{ref}$. After the base model generates the initial prediction $\hat{y}$, the policy network samples $N_s$ correction actions $a_j$ in parallel based on the state $[y^{ref}; \hat{y}]$. We then compute rewards and simultaneously update both the policy network and base model parameters (Total Loss $\mathcal{L}_ {total} = \gamma_1 MSE + \gamma_2 \mathcal{L}_ {RL}$). This design achieves end-to-end training, and since retrieval occurs only during training, there is **no additional overhead during inference**.
>
>
> **Q2: How can a subset of plausible historical examples be used to train the model, and how exactly are these examples selected?**
>
> We select the top-k most similar historical samples by calculating the Euclidean distance between the current input sequence and the sequences stored in our database. Addressing potential concerns about data leakage, we have strictly avoided this in our experimental design. Our retrieval database ($\mathcal{D}^R$) is constructed **exclusively from the training set data**. This ensures that during training, the model can only access "past" data and has absolutely no possibility of retrieving "future" data from the validation or test sets, thereby fundamentally preventing data leakage.
>
>
> **Q3: How much extra training costs are caused by your RAG and RL sampling procedure?**
>
> We have included **Table 6 and Table 7 in Appendix C.3** to quantify these costs. First, index construction is a one-time process and extremely fast; for example, it takes only 11.73 seconds on the largest Traffic dataset. Second, regarding dynamic training overhead, taking iTransformer on the ETTh1 dataset as an example, introducing SRRF (with sample count $N_s=4$) changes the training time per epoch from 3.53 seconds to 6.65 seconds. This investment in training time yields significant gains in predictive performance. Most importantly, SRRF is a **training-time only** enhancement; during inference, we do not perform any retrieval or RL computations, ensuring that the **inference overhead is strictly zero**, which guarantees efficiency in practical deployment.

---

### Meta-Review · Area_Chair_iucZ · 2026-01-06

**Summary:**

Reviewers raised consistent concerns about the practicality and scalability of the proposed framework, despite acknowledging the novelty of using RL as a corrective mechanism for spectral bias. While the rebuttal clarified the RAG–RL integration, corrected experimental issues, and added baselines and sensitivity analyses, the newly reported results also confirmed substantial training-time overhead, with costs growing linearly in the number of RL samples and becoming significant even at moderate settings. Several reviewers therefore questioned whether the demonstrated performance gains sufficiently justify the added training complexity and resource requirements, which ultimately informed the recommendation.

**Reviewer Concerns:**

The rebuttal addresses several reviewer concerns by clarifying the RAG–RL integration and correction mechanism, correcting experimental inconsistencies, adding missing baselines, expanding hyperparameter sensitivity analyses, and providing detailed measurements of computational overhead. However, the central concern regarding the practicality of the training-time cost remains outstanding, as the new results confirm substantial slowdowns with increasing RL sample counts, raising doubts about the overall cost–benefit trade-off of the approach in realistic training settings.

**Reviewer Scores:**

Overall, the rebuttal and revisions would likely lead to modest upward adjustments for some reviewers but not a decisive shift in consensus. Reviewers who were primarily concerned with clarity, missing baselines, and experimental inconsistencies would likely increase their scores slightly or maintain borderline-positive assessments. In contrast, reviewers who emphasized training-time overhead and practicality as fundamental issues would likely keep their scores largely unchanged, as these concerns persist despite improved measurement and reporting. As a result, the overall score distribution would remain mixed and centered around the borderline/reject threshold.

---

### Decision · Program_Chairs · 2026-01-26

Reject